environmental science/ecology/plant science

tropical forests, tree structure, above-ground biomass, destructive harvest, terrestrial lidar, allometry

**Author for correspondence:**
Andrew Burt
email: a.burt@ucl.ac.uk

# New insights into large tropical tree mass and structure from direct harvest and terrestrial lidar

Andrew Burt[1], Matheus Boni Vicari[1], Antonio C. L. da Costa[2], Ingrid Coughlin[3], Patrick Meir[3,4], Lucy Rowland[5] and Mathias Disney[1,6]

[1]Department of Geography, University College London, London, UK
[2]Instituto de Geociências, Universidade Federal do Pará, Belém, Brazil
[3]Research School of Biology, Australian National University, Canberra, Australia
[4]School of GeoSciences, University of Edinburgh, Edinburgh, UK
[5]College of Life and Environmental Sciences, University of Exeter, Exeter, UK
[6]NERC National Centre for Earth Observation (NCEO), Leicester, UK

AB, 0000-0002-4209-8101; MBV, 0000-0001-8841-4205;
ACLdC, 0000-0001-8140-4020; IC, 0000-0002-8541-2682;
PM, 0000-0002-2362-0398; LR, 0000-0002-0774-3216;
MD, 0000-0002-2407-4026

A large portion of the terrestrial vegetation carbon stock is stored in the above-ground biomass (AGB) of tropical forests, but the exact amount remains uncertain, partly owing to the lack of measurements. To date, accessible peer-reviewed data are available for just 10 large tropical trees in the Amazon that have been harvested and directly measured entirely via weighing. Here, we harvested four large tropical rainforest trees (stem diameter: 0.6–1.2 m, height: 30–46 m, AGB: 3960–18 584 kg) in intact old-growth forest in East Amazonia, and measured above-ground green mass, moisture content and woody tissue density. We first present rare ecological insights provided by these data, including unsystematic intra-tree variations in density, with both height and radius. We also found the majority of AGB was usually found in the crown, but varied from 42 to 62%. We then compare non-destructive approaches for estimating the AGB of these trees, using both classical allometry and new lidar-based methods. Terrestrial lidar point clouds were collected pre-harvest, on which we fitted cylinders to model woody structure, enabling retrieval of volume-derived AGB. Estimates from this approach were more accurate than allometric counterparts (mean tree-scale relative error: 3% versus 15%), and error decreased when up-scaling to the cumulative AGB of the four trees (1% versus 15%). Furthermore, while allometric error increased fourfold with tree size over the diameter range, lidar error

remained constant. This suggests error in these lidar-derived estimates is random and additive. Were these results transferable across forest scenes, terrestrial lidar methods would reduce uncertainty in stand-scale AGB estimates, and therefore advance our understanding of the role of tropical forests in the global carbon cycle.

## 1. Introduction

Forests in Amazonia are estimated to store in the region of 50–60 Pg of carbon in above-ground live vegetation [1–3], and are probably a small carbon sink, estimated at $0.25 \pm 0.3$ P g C yr$^{-1}$, but appear to be trending towards becoming a carbon source [4]. The uncertainty in these values has implications for policy assessment: commensurate uncertainty exists in the exact benefits provided by afforestation and reforestation programmes (e.g. REDD+ or carbon trading schemes), or likewise, the expediency of deforestation.

The cornerstone of any Amazon-scale estimate of carbon stocks are the networks of calibration sites; censused forest stands where the above-ground biomass (AGB; i.e. the mass of above-ground live plant matter at 0% moisture content) of each tree has been quantified (the carbon content of biomass varies, but is often observed between 45–50%) [5–7]. However, there is not a single hectare of tropical forest on Earth whose AGB has been directly measured. At these calibration sites then, measurements are replaced with estimates.

This is because directly measuring the AGB of a tree is resource intensive: it requires the tree to be harvested, and subsequent measurement of: (i) above-ground green mass via weighing, and (ii) the moisture content of this green mass. Indeed, we could find just 10 large tropical trees (stem diameter ≥0.6 m) in Amazonia whose above-ground green mass had been directly measured and recorded in the peer-reviewed English language literature, where data were enumerated [8]. This increased to 60 when direct weighing measurements of above-ground green mass were at least partially replaced with geometry-derived volume estimates [9–14].

In the absence of measurements, AGB must instead be estimated at these calibration sites using allometry [15]. Allometric models describe the correlations that exist between tree-scale AGB, and more readily measured tree variables (e.g. stem diameter). These models are constructed, usually via log-transformed linear regression, using calibration (in-sample) data collected from destructively harvested trees, where AGB was measured concurrent with these predictor variables. For tropical forests, calibration data are usually collected from across the region or population (i.e. Amazonia or pan-tropics), rather than more locally by either geography or taxa, owing to diversity [16]. Widely used pan-tropical allometric models also include additional predictor variables beyond stem diameter, such as tree height and wood density, partly to reduce the standard error of the regression (i.e. these variables are correlated with AGB), and partly to enable capture of systematic biogeographic variations in tree height and wood density [17–19].

Error in allometric estimates potentially arises from the selection, measurement and modelling of the calibration data, as well as from the measurement of the out-of-sample data (i.e. the predictor variables of the tree whose AGB is to be estimated). A trait of allometric error is that it is multiplicative: that is, it is well observed that variance in AGB is not constant across tree size (i.e. variance in AGB increases with increasing stem diameter and tree height; heteroscedasticity), therefore error itself increases with tree size [20]. At the tree-scale then, uncertainty in an allometric estimate of AGB is proportional to tree size, and it has been demonstrated that in tropical forests, it is possible for uncertainty to be larger than the estimate itself [21]. At the 1 ha stand-scale, random errors begin to average out, but it would still be reasonable to expect uncertainty to remain upwards of 40% [22]. Furthermore, unquantifiable systematic error probably increases these uncertainties further, regardless of scale, because of, among various reasons, imbalances in widely used calibration data, which are typically skewed toward smaller trees [23]. In absolute terms then, uncertainty in current state-of-the-art estimates of stand-scale AGB is potentially large, and relatively, a function of stand composition. Such uncertainties at the aforementioned calibration sites are of concern because they propagate directly through to larger-scale estimates of AGB and carbon stocks [24].

Recently, a new alternative method for estimating tree- and stand-scale AGB has been pioneered using terrestrial lidar data [25]. Modern terrestrial laser scanners enable capture of rich-but-unorganized point clouds that provide a millimetre-accurate three-dimensional sample of observable elements in forest scenes. Current hardware and data acquisition protocol enable high-quality data to be collected from, for example, a 1 ha tropical forest stand within a week of scanning [26]. Various tools and workflows have been developed to process these data: from the segmentation of point clouds representing individual trees, the classification of individual points as returns from either wood or leaf surfaces, to the construction of quantitative structural models (QSMs) via shape fitting, to explicitly describe woody

tree architecture [27–31]. Together, these methods enable retrieval of above-ground green woody volume from lidar data that, combined with a value of density, permit estimation of tree-scale AGB (note: these estimates of volume are derived from sampling external woody surfaces, so they are unable to account for either internal rot/cavities, or the contribution of leaf material to volume).

It is important at this point to briefly introduce the value of density required to convert these lidar-derived volumes into estimates of AGB. Throughout this paper, we use the rather cumbersome term 'above-ground basic woody tissue density', because the value is required to have the following three properties. First, the lidar data provide a three-dimensional sample of any particular tree in a green state (i.e. it is assumed cell walls are saturated, and lumina filled with water), so to estimate dry mass from green volume, a basic density (i.e. dry mass over green volume) is required. Second, this single value must account for the density of each above-ground woody tissue (i.e. periderm, phloem, cambium, xylem and pith) filling the volume, weighted by the relative abundance of each tissue. Third, this value must account for the intra-tree variations in the density of these tissues, which have been observed to vary with both radius and height [32,33].

The fundamental attraction of this lidar-based approach, over classical allometry, is that AGB estimates are not derived from calibration data collected from other, unrelated trees, but from explicit three-dimensional measurements of the particular tree itself. Additionally, the various processing methods (e.g. shape fitting) for estimating above-ground green woody volume are scale invariant, and therefore in an idealized scenario, there is no expectation for error in these estimates to be correlated with tree size; that is, error is additive. This is a potentially important differentiator between lidar- and allometric-based approaches, because it would mean that when these lidar-derived estimates are up-scaled, the error in stand-scale estimates would be independent of the stand and its composition. However, the question remains: what is the error, both random and systematic, in these estimates? While there are different approaches to answering this question, including simulation experiments, the gold standard is to compare estimates, $AGB_{est}$, with directly measured reference values, $AGB_{ref}$.

To date, several publications present studies validating $AGB_{est}$ retrieved from terrestrial lidar data via QSM-derived volume estimates. Calders *et al.* [34] collected both lidar and destructive measurements from 65 trees in *Eucalyptus* spp. open forest in Australia, and using their particular lidar data processing chain, found the coefficient of variation of the root mean square error in tree-scale $AGB_{est}$ to approximate 16%. Perhaps the most significant finding was the absence of a correlation between error and tree size, which provided the first empirical evidence to support the aforementioned hypothesis that error in these estimates is additive. For tropical forests, Momo *et al.* [35] acquired lidar and destructive data in Cameroon for 61 trees (mean stem diameter: 0.58 m, tree height: 34 m), and found the mean tree-scale relative error in $AGB_{est}$ approximated 23%. Gonzalez de Tanago *et al.* [36] also collected similar data in Guyana, Indonesia and Peru for 29 trees (mean stem diameter: 0.73 m), and found the coefficient of variation of the root mean square error in tree-scale $AGB_{est}$ to approximate 28%. In contrast to [34], error was observed to increase with tree size in both these studies. One possible explanation for this, and the overall increase in error, is the additional difficulty in acquiring high-quality lidar and destructive data in dense and remote tropical forests.

For example, in both [35,36], $AGB_{ref}$ could not always be obtained from direct measurements of green mass. Instead, for all trees [36], or for portions of large trees [35], mass was instead estimated from geometry: length and diameter measurements were converted into green volume estimates by assuming underlying tree structure could be represented by either a cylinder or conical frustum. Estimates of dry mass were then generated using an estimate of basic woody tissue density obtained from either the harvested trees themselves [35], or from a global database [36]. That is, despite the significant advances both studies achieved, both were also limited by the inability to determine how accurately $AGB_{ref}$ represented true AGB.

In this paper, we present, to our knowledge, the first validation of lidar-derived AGB estimates of tropical trees using reference values obtained entirely from direct measurement. We decided to focus the experiment on collecting high-quality and detailed measurements from a small sample of large trees, as opposed to collecting lower quality data from a larger sample. This approach provides the opportunity to generate a robust understanding of the minimum error that can be expected in lidar-derived AGB estimates, with currently available hardware and data processing methods. Overall, we harvested four large tropical rainforest trees (stem diameter range: 0.6–1.2 m, tree height range: 30–46 m) in an intact old-growth forest stand in East Amazonia, and weighed their above-ground green mass. We also gathered multiple discs from the stem and crown of each tree, on which moisture content and woody tissue density were measured. Terrestrial lidar measurements were collected from the four trees pre-harvest using a high-performance laser scanner after neighbouring trees and

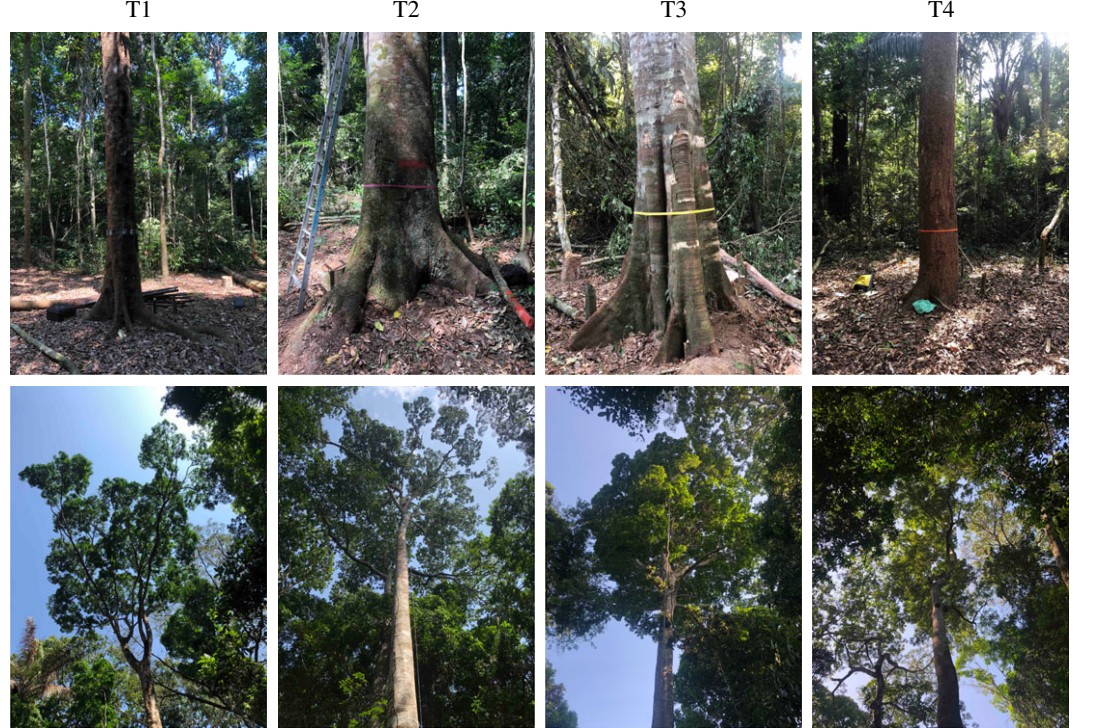

**Figure 1.** Photographs of the stem and crown of each harvested tree. Left to right: T1 (*Inga alba* (Sw.) Willd.), T2 (*Hymenaea courbaril* L.), T3 (*Tachigali paniculata* var. *alba* (Ducke) Dwyer) and T4 (*Trattinnickia burserifolia* Mart.).

understory were cleared to minimize occlusion. Tree-scale $AGB_{est}$ was then retrieved from these data using a replicable and open-source processing chain. Crucially, these methods rely solely on the lidar data themselves for calibration (i.e. $AGB_{est}$ was not *a priori* informed by $AGB_{ref}$).

In the results section, we first present the reference data from the destructive measurements. This includes rare and interesting ecological insights provided by these data, including, for each tree, the moisture content and the distribution of mass between stem and crown, as well as the intra-tree variations in basic woody tissue density. We then present the lidar data, and quantify the error in $AGB_{est}$ retrieved from these data. We show how these errors compare with those arising from allometry. We then discuss the wider implications of these results for improving tropical forest tree- and stand-scale AGB estimates and related ecological understanding.

# 2. Methods

Destructive and non-destructive data were collected from four trees (designated T1–T4, pictured in figure 1) in Floresta Nacional de Caxiuanã, Pará, Brazil, during September and October 2018. These data are open-access and distributed under the terms of the Creative Commons Attribution 4.0 International Public License (CC BY 4.0). Persistent identifiers for these data are available in the data accessibility section.

## 2.1. Site metadata

The approximate coordinates of the site in the WGS-84 datum are −1.798°, −51.435°. The site is classified as moist, terra firme, pre-quaternary, lowland, mixed species, old-growth, intact tropical forest. Mean annual rainfall is 2000–2500 mm, with a dry season between June and November. Soils at a nearby long-term through-fall exclusion experiment (approx. 7 km from the site) are yellow oxisol, composed of 75–83% sand, 12–19% clay and 6–10% silt [37,38].

## 2.2. Tree selection

We determined tree selection should principally ensure there was some range in the values of stem diameter and tree height. Other considerations influencing selection was a bias towards those that appeared healthy and with a complete crown, felling (in particular, that no large neighbouring trees would affect felling

operations), and also permission from the land owners. The four selected trees were identified to species, and foliage/fruit samples retained to confirm taxonomic identity at the Museu Paraense Emílio Goeldi, Belem, Pará, Brazil. Nearby vegetation surrounding each tree was removed, including complete clearing of small bushes and saplings, to enable detailed non-destructive and destructive activities.

## 2.3. Non-destructive measurements

Stem diameter was measured using a circumference/diameter tape. Measurement was made at either 1.3 m above-ground (T4), or 0.5 m above-buttress (T1–T3). Lidar data were collected using a RIEGL VZ-400 terrestrial laser scanner. A minimum of 16 scans (upright and tilt) were acquired from eight scan positions around each tree, with a variable distance between scanner and tree. This arrangement provided a 45° sampling arc around each tree, and a complete sample of the scene from each position. The angular step between sequentially fired pulses was 0.04°. The laser pulse has a wavelength of 1550 nm, a beam divergence of 0.35 mrad, and the diameter of the footprint at emission is 7 mm. The instrument was in 'High-Speed Mode' (pulse repetition rate: 300 kHz), 'Near Range Activation' was off (minimum measurement range: 1.5 m), and waveforms were not stored.

## 2.4. Destructive measurements

The various field and laboratory measurements are described below. Included in the destructive dataset are photographs illustrating these measurements.

### 2.4.1. Field measurements

The location, height and above-ground green mass of the four trees were measured in the field, and multiple discs were also collected from each tree. Trees were cut at a height of approximately 1 m above-ground, and felled onto tarpaulin. A STIHL MS 650 chainsaw with a 20″ bar length and 13/64″ chain loop was used for cutting. Tree height (including stump) was then measured with a surveyor's tape measure, and GPS data were acquired from the centre of the stump using a Garmin GPSMAP 64st.

Two Adam LHS 500 crane scales (capacity: 500 kg, division: 0.1 kg) were suspended from nearby trees to measure green mass. Calibration certificates are included in the destructive dataset certifying both instruments conformed (pre-delivery) to various tests of repeatability, linearity and hysteresis. Post-campaign, the consistency of measurements between both scales was tested using a variety of different masses, and measurements were also compared against the laboratory balance with internal calibration weights described in the following subsection.

Green mass was assigned to either the stem or crown pool, where the stem was defined as the bole from flush with the ground up to the first fork. The stem was cut into manageable sections, the length and diameter of each section was measured using a circumference/diameter tape (lengths varied from 0.15–0.54 m for T2 and T4, respectively), and then weighed. Remaining stump material was cut flush with the ground and weighed. Large branches were cut into manageable sections, and fine branching and leaf material were collected in large sacks, and weighed. These green mass measurements commenced immediately post-felling, but two, four, four and two days were required to complete measurement of T1–T4, respectively.

Discs of approximately 50 mm thickness were collected from the stem and crown of each tree. Four discs were collected from the stem: at 1.3 m above-ground, and at 25%, 50% and 75% the length of the stem. Up to three discs were also collected from the mid-points of first, second and third-order branches. A minimum of 11 discs were collected per tree. Figure 2 shows the discs collected from T4.

### 2.4.2. Laboratory measurements

Discs were reduced to subsamples using a consistent approach: for any particular disc, they were taken from along the major axis, as guided by parallel chords straddling above and below the length of this axis, each with approximate dimensions of $150 \times 50 \times 50$ mm. That is, each set of subsamples covered the length of their respective disc (i.e. included periderm, phloem, cambium, xylem and pith tissues). Figure 2 also shows the sets of subsamples cut from the discs of T4.

The mass and volume of each subsample was measured while in both a green and dry state. These measurements were made using an Adam NBL4602i Nimbus Precision balance (capacity: 4600 g, division: 0.01 g). This balance contains internal calibration weights, and a calibration certificate is available

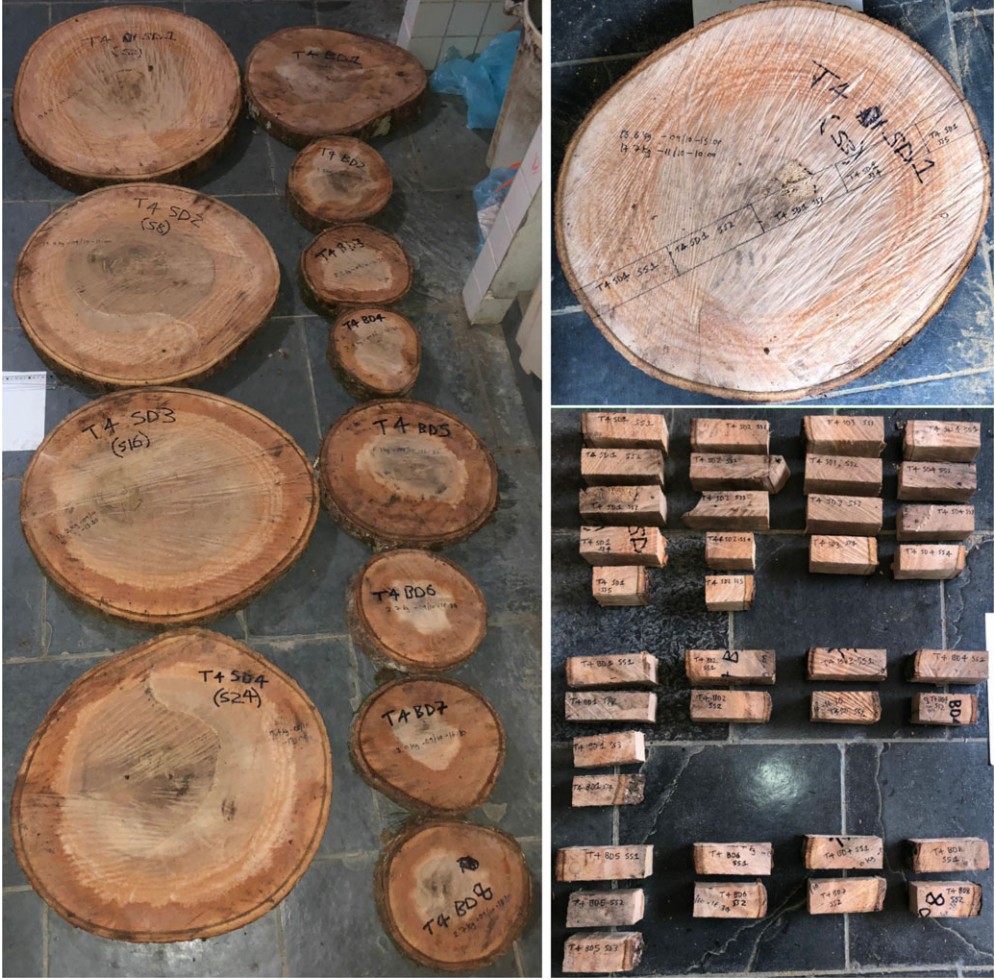

**Figure 2.** The photograph on the left shows the discs collected from T4. Four were acquired from the stem, at 1.3 m above-ground, and at 25%, 50% and 75% the length of the stem. Eight were taken from the mid-points of first-, second- and third-order branches. Each disc was subsequently reduced to a set of subsamples using a consistent approach: for any particular disc they were taken from along the major axis, as guided by parallel chords straddling above and below the length of this axis (upper right photograph). The lower right photograph shows all subsamples from T4. It is noted that each set of subsamples cover the full length of their corresponding disc, and therefore each set includes periderm, phloem, cambium, xylem and pith tissues.

in the harvest dataset certifying this instrument conformed (pre-delivery) to external tests of repeatability, eccentric loading, linearity and hysteresis. Volume measurements were made using the balance via Archimedes' principle. Subsamples were considered in a green state after soaking for 48 h. They were considered in a dry state once a constant mass had been attained while drying in an oven at 105°C.

In the original campaign, the green volume of subsamples was not measured (necessary for calculating basic woody tissue density). We returned to the harvest site in October 2019 and collected new discs and subsamples from the wood piles of each tree using the methods described above. The integrity of the wood was verified by comparing measurements of woody tissue green-to-dry mass ratio and dry woody tissue density. The green volume of the original subsamples was retrospectively calculated using the mean woody tissue green-to-dry volume ratio from these new subsamples.

## 2.5. Above-ground biomass

### 2.5.1. Woody tissue green-to-dry ratios and density

Above-ground woody tissue green-to-dry mass and volume ratios were estimated for each tree from measurements on the subsamples using a mass-weighted approach. These two variables were calculated by weighting the mean from the subsamples in each pool (stem and crown), with the green mass in each pool. Above-ground basic woody tissue density was estimated using three approaches:

(i) mass-weighted; (ii) stem, which was calculated from the mean of the two outer subsamples taken from the stem disc at 1.3 m above-ground; and (iii) literature, where values were taken from the Global Wood Density Database [39], whereby the value considered was the mean of matched species-level entries. The second approach was intended to mimic values acquired via increment boring (i.e. a regular wood density measurement), and the third approach was considered because it is widely applied in non-destructive settings.

### 2.5.2. Reference values

$AGB_{ref}$ was calculated from measured above-ground green mass and the mass-weighted estimate of above-ground woody tissue green-to-dry mass ratio. Mass lost in swarf from chainsaw cuts was partially accounted for using a volume-derived approach (stem only). A green cut volume was estimated for each of the manageable stem sections by assuming it could be represented by a cylinder. An average cut width of 8.4 mm (approx. 60% thicker than the chain loop) was estimated from multiple caliper measurements. Lost dry mass was estimated from these green volumes using the mass-weighted estimates of above-ground green woody tissue density and above-ground woody tissue green-to-dry mass ratio. These corrections increased $AGB_{ref}$ by 0.7%, 1.8%, 0.7% and 0.7% for T1–T4, respectively.

### 2.5.3. Lidar-derived estimates

Estimates of above-ground green woody volume were retrieved from the lidar data using the software described below. With the exception of the initial step, these tools are open-source, and persistent identifiers for the source code are available in the data accessibility section. Individual lidar scans were registered onto a common coordinate system using RiScan Pro (v. 2.7.0) [40]. The point clouds representing the individual trees were extracted from the larger-area point clouds using TreeSEG (v. 0.2.0) [28]. Returns from leafy surfaces were segmented from these tree-level point clouds after wood-leaf classification using TLSeparation (v. 1.2.1.5) [27]. Points from buttresses were then manually removed using CloudCompare (v. 2.10.3) [41]. QSMs were constructed from these point clouds using TreeQSM (v. 2.3.2) [31]. Input parameters to TreeQSM were automatically selected using optQSM (v. 0.1.0). $AGB_{est}$ was calculated from estimated above-ground green woody volume and the mass-weighted estimate of above-ground basic woody tissue density.

### 2.5.4. Allometric-derived estimates

$AGB_{est}$ was calculated using the two-parameter model described in Chave *et al.* [11] (eqn (4), three predictor variables: stem diameter, $D$ (cm), tree height, $H$ (m) and basic wood density, $\rho$ (g cm$^{-1}$)). This model is constructed from measurements collected on 4004 destructively harvested trees from across the tropics, and takes the form:

$$AGB_{est} = 0.067(D^2H\rho)^{0.976}, \tag{2.1}$$

where here, stem diameter was measured pre-harvest using a circumference/diameter tape at 1.3 m above-ground (T4) or 0.5 m above-buttress (T1–T3), tree height was measured post-harvest using a surveyor's tape measure, and basic wood density was represented by the mass-weighted estimate of above-ground basic woody tissue density. The electronic supplementary material, appendix A considers our decision to select this particular model to represent the allometric approach.

### 2.5.5. Estimate performance

To describe the performance of lidar- and allometric-derived $AGB_{est}$, we use the terms random error, systematic error, total error, precision, trueness, accuracy, bias and uncertainty. For the avoidance of doubt, these terms are used within the International Organization for Standardization 5725 and the International Bureau of Weights and Measures definitions [42]. Error, $\varepsilon$, in $AGB_{est}$ is defined as

$$\varepsilon = AGB_{est} - AGB_{ref.} \tag{2.2}$$

Relative error is defined as

$$RE = \frac{|\varepsilon|}{AGB_{ref}}. \tag{2.3}$$

Mean tree-scale relative error is defined as

$$\mathrm{MRE} = \frac{1}{n}\sum_{i=1}^{n}\mathrm{RE}_i. \tag{2.4}$$

Mean error is used to quantitatively express trueness:

$$\mathrm{ME} = \frac{1}{n}\sum_{i=1}^{n}\varepsilon_i. \tag{2.5}$$

The standard deviation of the error is used to quantitatively express precision:

$$\mathrm{STDEV} = \sqrt{\frac{1}{n}\sum_{i=1}^{n}(\varepsilon_i - \bar{\varepsilon})^2}. \tag{2.6}$$

Root mean square error is used to quantitatively express accuracy:

$$\mathrm{RMSE} = \sqrt{\frac{1}{n}\sum_{i=1}^{n}\varepsilon_i^2}. \tag{2.7}$$

# 3. Results

## 3.1. Ecological insights from the harvest measurements

Directly measured above-ground green mass of the four trees ranged from 6808 to 29 511 kg, totalling 59 005 kg (table 1). The disc-derived mass-weighted estimates of above-ground woody tissue green-to-dry mass ratio ranged from 1.62 to 1.73, implying that in the region of 40% of this green mass was water. $AGB_{ref}$, calculated from the above values, ranged from 3960 to 18 584 kg, totalling 36 458 kg. The proportion of $AGB_{ref}$ distributed between the stem and crown varied between trees, with the crown mass ratio ranging from 0.42 to 0.62, but for three out of four, the majority was found in the crown (figure 3).

The mass-weighted estimates of above-ground basic woody tissue density ranged from 568 to 769 kg m$^{-3}$ for the four trees. Values generated from the other direct approach using the discs (stem), which were intended to mimic measurements acquired via increment boring, were consistent with mass-weighted counterparts, with a maximum percentage difference of 2% (table 1 and figure 4). Non-direct values obtained from the literature were less consistent: for T1 and T2 they were close, but were significantly lower than mass-weighted estimates in both T3 and T4, with a percentage difference of 3%, 3%, 14% and 20% for T1–T4, respectively.

Beneath these whole-tree values were substantial and unsystematic intra-tree variations in basic woody tissue density (figure 4). Overall, density decreased with height in T2, increased in T3 and T4, and remained largely invariant in T1. A pronounced decrease was also observed towards the centre of each stem disc in T3, which was not seen in the other trees. The maximum difference between measurements on the subsamples collected throughout each tree was 40 kg m$^{-3}$, 79 kg m$^{-3}$, 166 kg m$^{-3}$ and 156 kg m$^{-3}$ for T1–T4, respectively.

## 3.2. Non-destructive above-ground biomass estimates

The downsampled terrestrial lidar data comprised a minimum 2.6 million points per tree (figure 3). The QSMs constructed from the woody point clouds are shown in figure 5. $AGB_{est}$, derived from the volume of these QSMs, had a mean tree-scale relative error of 3% (table 2). Relative error in up-scaled $AGB_{est}$ (i.e. the error in the cumulative $AGB_{est}$ of the four trees) was 1%.

The mean tree-scale relative error in allometric-derived $AGB_{est}$ was 15%, and the relative error in up-scaled $AGB_{est}$ was also 15%. Overall then, the lidar methods returned more accurate predictions than allometry (table 2). Precision-wise (random error), the standard deviation of the error in lidar-derived estimates was considerably smaller: 185 kg versus 2523 kg. These estimates were also more true (systematic error), but did tend to be underestimates (mean error: −70 kg versus 1398 kg).

The lidar methods also accurately estimated the allocation of AGB intra-tree, as illustrated in figure 6. This compares the distributions of AGB between the stem and crown, for both these estimates and the reference measurements. The distributions are similar: 48%, 58%, 38% and 41% of harvest-derived

**Table 1.** Variables of the four harvested trees. (The approach to each measurement is described in the methods section. It is noted that conversion of above-ground green mass to above-ground dry mass via above-ground woody tissue green-to-dry mass ratio does not yield the reported value. This is because above-ground dry mass includes a correction to partially compensate for mass lost in swarf from chainsaw cuts. These corrections were 27.7 kg, 330.2 kg, 58.1 kg and 41.0 kg for T1–T4, respectively.)

| ID | species | coords (°) | $D$ (m) | $H$ (m) | above-ground green mass (kg) | above-ground dry mass ($AGB_{ref}$) (kg) | above-ground basic woody tissue density (kg m$^{-3}$) | | | above-ground woody tissue green-to-dry ratio | |
| | | | | | | | mass-weighted | stem | literature | mass | volume |
| --- | --- | --- | --- | --- | --- | --- | --- | --- | --- | --- | --- |
| T1 | Inga alba (Sw.) Willd. | − 1.79851 − 51.43463 | 0.647 | 29.8 | 6808.2 | 3960.1 | 567.6 | 574.6 | 586.1 | 1.731 | 1.169 |
| T2 | Hymenaea courbaril L. | − 1.79832 − 51.43479 | 1.179 | 46.2 | 29511.0 | 18584.2 | 768.9 | 786.7 | 792.4 | 1.617 | 1.126 |
| T3 | Tachigali paniculata var. alba (Ducke) Dwyer | − 1.79914 − 51.43451 | 0.905 | 34.9 | 13697.5 | 8392.6 | 639.0 | 655.1 | 554.5 | 1.643 | 1.137 |
| T4 | Trattinnickia burserifolia Mart. | − 1.79550 − 51.43433 | 0.697 | 35.2 | 8988.6 | 5521.1 | 567.0 | 561.9 | 460.0 | 1.640 | 1.113 |

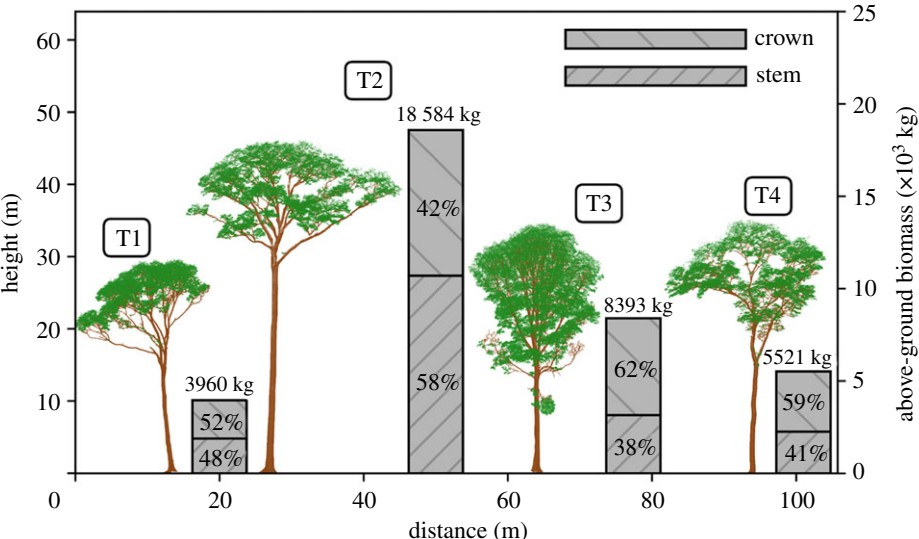

**Figure 3.** The terrestrial lidar point clouds collected from the four trees prior to harvest. Shown to scale, left to right: T1–T4. These clouds were segmented from the larger-area point clouds using TREESEG [28], and individual points were classified as returns from wood (brown) and leaf (green) surfaces using TLSEPARATION [27]. The bars on the secondary y-axis provide the harvest-derived reference values of above-ground biomass (AGB$_{ref}$). Also shown in these bars are the distributions of AGB$_{ref}$ between the stem and crown.

AGB$_{ref}$ was found in the stems of T1–T4, respectively, while 48%, 62%, 42% and 47% of lidar-derived AGB$_{est}$ was, respectively, allocated to each stem.

Finally, the approach to estimating above-ground basic woody tissue density was an important determinant of error (figure 7). For the lidar methods, the values derived from direct disc measurements provided estimates with the smallest error: mean tree-scale relative error in AGB$_{est}$ using the mass-weighted, stem and literature estimates was 3%, 4% and 9%, respectively. This was less marked for the allometric estimates, with mean tree-scale relative error remaining largely constant across the three approaches (15%, 16% and 18%, respectively).

# 4. Discussion

The destructive measurements provide a rare insight into the considerable variability in large tropical tree structure. The distribution of AGB between the stem and crown of the four trees varied widely, with the crown mass ratio ranging from 0.42 to 0.62 (figure 3). Above-ground basic woody tissue density was estimated (via the mass-weighted approach) to range from 567 to 769 kg m$^{-3}$ across the four trees. Density was also observed to vary substantially intra-tree, with both height and radius, and further, these variations were not systematic between trees (figure 4). Less variable were above-ground woody tissue green-to-dry mass and volume ratios: the moisture content of green mass varied from 39 to 42%, and green woody tissues shrank by 10–14% after drying (table 1).

The harvest measurements found AGB$_{ref}$ of the trees to total 36 458 kg. Non-destructive AGB$_{est}$ from the lidar- and allometric-derived methods totalled 36 178 kg and 42 051 kg, respectively (table 2). The mean tree-scale relative error was 3% and 15%, respectively. In addition to smaller error, lidar-derived estimates had two further advantages. First, error did not increase with increasing tree size (i.e. stem diameter and tree height), and second, error reduced when up-scaling to the cumulative of the four trees.

## 4.1. Lidar-based methods can accurately estimate above-ground biomass

It was noted in the introduction that error in pan-tropical tree-scale AGB$_{est}$ from allometry is multiplicative (i.e. error is proportional to tree size), and that it would not be unreasonable to expect

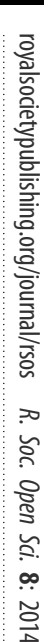

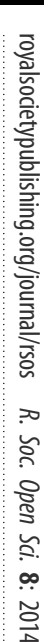

**Figure 4.** Intra-tree variations in basic woody tissue density with height, for the four harvested trees (T1–T4). Discs were collected from the stem of each tree at 1.3 m above-ground, and at 25%, 50% and 75% along the length of the stem (SD1-SD4, respectively). Discs were also taken from the midpoints of first-, second- and third-order branches (BO1-BO3, respectively). A minimum of 11 discs were collected per tree. These discs were reduced to subsamples using a consistent approach (figure 2), such that any given set of subsamples covered the full length of the major axis of each disc. Each datum represents a measurement of basic woody tissue density on a subsample: crosses represent outer subsamples (i.e. they included periderm, phloem and cambium tissues), and dots represent inner subsamples. The mean density at each disc location is marked by a grey square, and the solid grey interpolation line links these averages throughout each tree. The horizontal dashed grey lines show the range of density values per location, and each plot has an overall range of 350 kg m$^{-3}$ for comparative purposes. The vertical lines represent the estimates of above-ground basic woody tissue density derived from the three considered approaches: mass-weighted (red), stem (green) and literature (blue).

uncertainty from random error to remain upwards of 40% at the 1 ha stand-scale. Uncertainty then, in current best estimates of stand-scale AGB, is, in absolute terms, potentially large, and relatively, a function of plot composition. These uncertainties form the current minimum uncertainty in larger-scale AGB estimates (e.g. Amazonia), because such estimates are themselves invariably reliant upon calibration from forest stands characterized by allometry. The results from this study suggest terrestrial lidar methods have the potential to reduce these uncertainties by more than an order of magnitude, which would enable the production of significantly more accurate estimates of larger-scale carbon stocks.

Overall, lidar-derived AGB$_{est}$ provided a 15-fold increase in accuracy over allometric counterparts (RMSE difference, table 2). There were two important characteristics that further differentiated these estimates: (i) error in lidar-derived AGB$_{est}$ did not increase with tree size: tree-scale relative error in

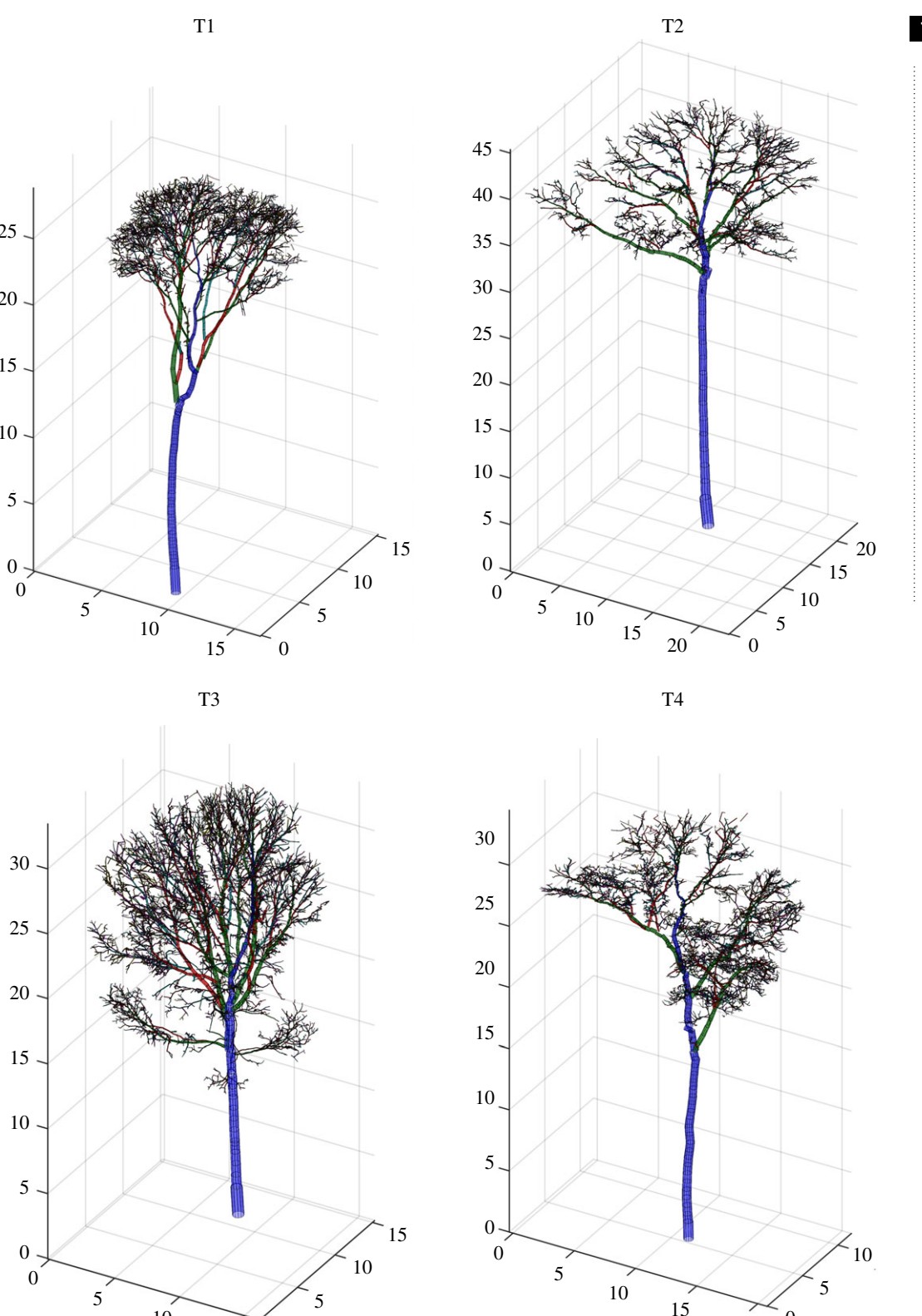

**Figure 5.** Three-dimensional views of the quantitative structural models representing the four harvested trees. All dimensions in metres. These cylinder models were constructed from the woody point clouds shown in figure 3 using TREEQSM [31]. Colours correspond to branching order (i.e. blue, green and red represent stem, first- and second-order branches, respectively).

the smallest and largest tree was 7% and 1%, respectively (versus 8% and 31%); and (ii) error in lidar-derived $AGB_{est}$ reduced when up-scaling from the tree-scale, to the cumulative of the four trees: mean tree-scale relative error was 3%, and the relative error in cumulative $AGB_{est}$ was 1% (versus 15% and

**Table 2.** The performance of lidar- and allometric-derived estimates of above-ground biomass ($AGB_{est}$). (Estimates from both approaches were calculated using the mass-weighted estimate of above-ground basic woody tissue density, and compared with the harvest-derived reference values ($AGB_{ref}$). The upper table reports the error in these estimates. Mean relative error is the average of tree-scale relative errors. Summed values refer to the cumulative AGB of the four trees. The lower table quantifies the trueness, precision and accuracy of these estimates.)

| ID | $AGB_{ref}$ (kg) | terrestrial lidar | | | pan-tropical allometry | | |
|---|---|---|---|---|---|---|---|
| | | $AGB_{est}$ (kg) | error (kg) | relative error (%) | $AGB_{est}$ (kg) | error (kg) | relative error (%) |
| T1 | 3960.1 | 3673.8 | −286.3 | 7.2 | 3644.9 | −315.2 | 8.0 |
| T2 | 18584.2 | 18414.5 | −169.7 | 0.9 | 24261.0 | 5676.8 | 30.5 |
| T3 | 8392.6 | 8604.6 | 212.0 | 2.5 | 9190.9 | 798.3 | 9.5 |
| T4 | 5521.1 | 5485.1 | −36.0 | 0.7 | 4953.7 | −567.4 | 10.3 |
| mean | — | — | — | 2.8 | — | — | 14.6 |
| summed | 36458.0 | 36177.9 | −280.1 | 0.8 | 42050.5 | 5592.5 | 15.3 |

| error type | performance characteristic | quantitative metric | estimation method | |
|---|---|---|---|---|
| | | | terrestrial lidar | pan-tropical allometry |
| systematic | trueness | mean error (kg) | −70.0 | 1398.1 |
| random | precision | standard deviation of the error (kg) | 185.3 | 2523.2 |
| total | accuracy | root mean square error (kg) | 191.1 | 2884.7 |

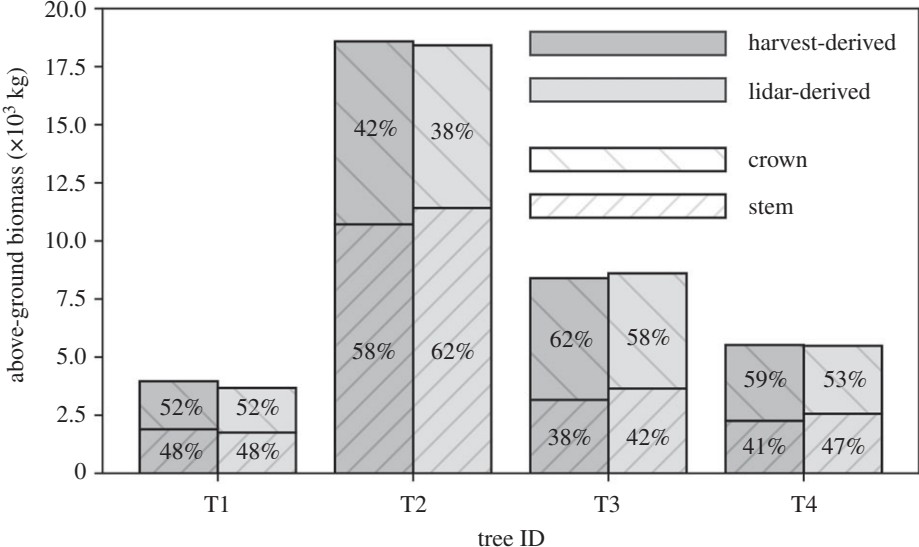

**Figure 6.** Comparison between the reference harvest-derived measurements of above-ground biomass (AGB), and the lidar-derived estimates. Estimates were calculated using the above-ground green woody volumes provided by the quantitative structural models shown in figure 5, and the corresponding mass-weighted estimates of above-ground woody tissue density. Also shown for both approaches, are the distributions of AGB between the stem and crown.

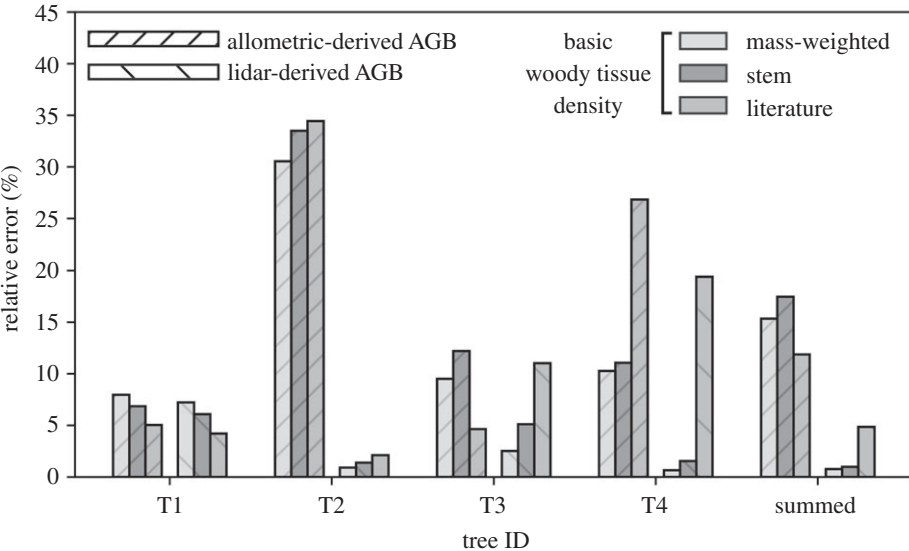

**Figure 7.** The impact of different approaches to estimating above-ground basic woody tissue density on the relative error in allometric- and lidar-derived estimates of above-ground biomass ($AGB_{est}$). Here, the mass-weighted, stem and literature values reported in table 1 were used in the calculation of $AGB_{est}$. Summed values refer to the cumulative AGB of the four trees.

15%). Combined, these results suggest that error in these particular lidar-derived estimates was additive (i.e. errors are not correlated with tree size) and random (i.e. errors are not correlated with one another). If these characteristics, derived from high-quality data collected on a small sample of four trees, were transferable across forest scenes, then error in lidar-derived stand-scale $AGB_{est}$ would be independent of the size of the trees comprising the stand, and reduce as the number of trees inside the stand increases.

Interestingly, the lidar-derived estimates presented here were also substantially more accurate than those from other studies, where estimates were accompanied by reference harvest data. As discussed in the introduction, two previous experiments have been conducted in tropical forests [35,36], and in both, it was found mean tree-scale relative error in lidar-derived $AGB_{est}$ exceeded 20% (versus 3% observed here). Because these lidar-derived estimates are generated from explicit three-dimensional models of tree structure, errors arise from two sources: the estimation of volume, and the estimation

of density. Discussed below, we believe recent advances in processing methods have significantly reduced error in estimating volume. However, knowledge gaps remain, particularly in the estimation of density. These knowledge gaps will require filling before lidar-derived estimates are considered both consistently true and precise.

### 4.1.1. Retrieving above-ground green woody volume from lidar point clouds

A laser scanner estimates the range to targets illuminated by the laser pulse, from timing measurements on the returned radiation. A sequence of methods are required to go from collecting these range estimates, to estimating volumes. Many sources of error exist in these methods, which can be broadly classified into two categories: those arising from data acquisition, and those arising from data processing.

Occlusion is an important source of error in the first category (i.e. tree components obscured by foreground objects). The impacts of occlusion on data quality are driven by multiple factors including the complexity of the scene, increasing distance between sequentially fired pulses with increasing range, instrument characteristics (e.g. beam divergence), and the user-defined sampling protocol. Other sources of error in the first category include the instrument itself (e.g. ranging accuracy), environmental impacts including wind and rain, and the inability to collect data from inside the tree (e.g. information on internal cavities). Sources of error in the second category are specific to the particular processing chain, but here, would include those arising from: (i) the registration of individual scans onto a common coordinate system, (ii) the segmentation of individual trees from the larger-area point cloud, (iii) the classification of points as returns from wood or leaf surfaces, (iv) the manual removal of points from buttresses, and (v) the approach to constructing QSMs (e.g. cylinders were used here to model woody structure).

It is impossible here to attribute the observed differences in $AGB_{est}$ and $AGB_{ref}$ to any of these sources. Indeed, attributing error to either top-level error source (i.e. the estimate of volume or density) is difficult. That is, we cannot explicitly explain the error because we do not have control over these individual sources, and because these errors are often related in complex ways (e.g. occlusion will affect the goodness of a cylinder fit). However, we have demonstrated that given occlusion minimized high-quality terrestrial lidar data, it can be expected that current open-source processing tools are capable of estimating large tropical tree AGB to within a few percent of reference measurements, despite these potential sources of error in volume estimation.

We think the recent development of wood-leaf separation algorithms has been an important step forward in enabling this [27,30]. This is illustrated in the electronic supplementary material, appendix B, where it is demonstrated that when returns from leafy surfaces are allowed to interfere in woody reconstruction, $AGB_{est}$ is substantially overestimated, with the relative error in cumulative $AGB_{est}$ increasing to upwards of 40%. This is further corroborated by figure 6, which demonstrates that the lidar methods were able to accurately resolve the allocation of AGB between the stem and crown of each tree. That is, $AGB_{est}$ did not accidentally agree with $AGB_{ref}$ because of underestimation in the stem and overestimation in the crown, but because of close agreement in both pools. We recommend then, that for the particular processing methods used here, removing leaf returns from point clouds is a critical step in accurately retrieving $AGB_{est}$ from occlusion minimized high-quality terrestrial lidar data collected in evergreen forests, or from so-called 'leaf-on' point clouds.

### 4.1.2. The more tricky issue of density

The other top-level source of error in lidar-derived $AGB_{est}$ is in the estimation of density. As described in the introduction, if we were to assume an estimate of a tree's above-ground green woody volume is error free, then to convert this into an unbiased estimate of AGB (assuming the contribution by leaf material to AGB is negligible), the above-ground basic woody tissue density is required. We have intentionally used this specific term throughout the paper because the value is required to: (i) convert a green volume into a dry mass, (ii) account for the density of each woody tissue filling the volume, by the relative abundance of each tissue, and (iii) account for the variations in the density of these tissues throughout the tree.

However, this variable is largely immeasurable because it requires above-ground woody weighing and water displacement. Here, the mass-weighted approach most closely resembled its measurement (i.e. weighting the mean basic woody tissue density from the disc subsamples in each pool, by the green mass in each pool). This mass-weighted approach illustrates how the more indirect methods for estimating above-ground basic woody tissue density introduce error into lidar-derived $AGB_{est}$ (figure 7). For example, the approach intended to mimic measurements on cores acquired from increment boring

(i.e. stem: calculated from the mean woody tissue density of the two outer subsamples taken from the stem disc collected at 1.3 m above-ground), increased mean tree-scale relative error from 3%, to 4%. Similarly, when we used values from the Global Wood Density Database [39], error increased to 9%. The point then, is that whenever non-destructive lidar-derived AGB estimates are required, the only practicable approaches to estimating above-ground basic woody tissue density are to use values obtained from increment boring or from the literature. The question is, are there any reliable methods to account for error arising from the inability to directly measure above-ground basic woody tissue density?

One approach is convert values obtained from either of these methods via some correction factor. Sagang *et al.* [33] and Momo *et al.* [43] have recently explored this approach for tropical forests in Central Africa. In these studies, data from destructive measurements ($n = 130$ and 822 trees, respectively) were used to construct models to convert literature- or increment boring-based values to more closely mirror above-ground basic woody tissue density. Momo *et al.* [43] also captured terrestrial lidar data for a subset of these harvested trees ($n = 58$), and demonstrated error in lidar-derived $AGB_{est}$ reduced when these corrective models were applied.

It would be unreasonable to apply the published corrective models to the data collected here, given the substantial variations observed within and between tropical regions [19]. It is however perhaps worth noting the top-level consequence had they been applied: our stem- and literature-derived estimates of above-ground basic woody tissue density would have been pushed downwards by approximately 21% and 36%, respectively, and therefore significantly below the mass-weighted estimates (figure 4). Here then, this behaviour would have greatly increased tree-scale relative error in lidar-derived $AGB_{est}$ for all four trees.

This is not to suggest such models are necessarily inappropriate at a larger-scale, (e.g. an equivalent model constructed from Amazonia data could be directionally correct), but at the out-of-sample tree- or stand-scale, it is impossible to know whether their application would have a beneficial or detrimental impact on error in $AGB_{est}$. Another important consideration for these models in the context of Amazonia is the hyperdominance of certain species [44]. That is, it is not necessarily sufficient for these models to accurately correct for the majority of species in the Amazon, if a large portion of Amazon-scale AGB is stored in a relatively small number of key species.

We would suggest then, there is currently no simple approach to account for error in lidar-derived $AGB_{est}$ arising from the inability to measure above-ground basic woody tissue density. In the longer-term, hopefully new non-destructive estimation methods will be pioneered, with interesting research ongoing using genomics [45]. However, in the meantime, our recommendation from the analysis of the data collected here, is that when viable, density estimates acquired from direct measurements on the stems of trees are probably superior to those taken from the literature (mean tree-scale relative error of 4% and 9%, respectively).

### 4.1.3. Improving uncertainty quantification

It is important to be able to assign meaningful uncertainty intervals to any non-destructive estimate of AGB, whether allometric- or lidar-derived, and whether at tree- or stand-scale. That is, when such methods are applied to a tree or stand where destructive validation measurements are unavailable, what is the uncertainty in subsequent estimates?

As discussed above, our analysis of the data collected from this sample of four large tropical trees identified: (i) error in lidar-derived $AGB_{est}$ did not increase with increasing tree size, and (ii) error in $AGB_{est}$ reduced when up-scaling from tree-scale, to the cumulative of the four trees. It would therefore be reasonable to suggest the following hypothesis: error in terrestrial lidar-derived tree-scale $AGB_{est}$ is additive and random. If this were provisionally confirmed, then to assign meaningful uncertainty intervals to out-of-sample tree- and stand-scale $AGB_{est}$ would only require a robust understanding of the variance in the error. To test this hypothesis, further destructive validation data similar to those collected here are required. In particular, it would be necessary to have some confidence that these new data attempted to capture the bounds of variability in above-ground volume and woody tissue density. That is, these data would ideally be collected from a range of different forest types, species, size classes and crown forms.

If such data were to be collected, then we think priority should be placed on two further considerations. First, on understanding the implications of lidar data quality on error. That is, how scene complexity, instrument and sampling patterns change random error variance and/or introduce systematic error. We note that in this experiment, the lidar data were collected after neighbouring

vegetation surrounding each tree was removed, meaning data quality is probably higher than equivalent data collected in a non-destructive setting.

Second, the reference measurements should be collected from direct measurement of green mass (i.e. not indirectly estimated via geometry measurements), to ensure reference values are as close to true as possible. The measurements collected in this experiment illustrate how important this point is: during the harvest, each stem was cut into manageable sections, on which length and diameter measurements were made, permitting estimation of geometry-derived (Smalian's formula) green stem mass via the mass-weighted estimate of above-ground green woody tissue density. A percentage error of $-1\%$, $7\%$, $11\%$ and $11\%$ was observed when these estimates were compared with direct weighing measurements for T1–T4, respectively. These are large errors, and their presence in reference data would lead to spurious interpretations of the error in lidar- or allometric-derived $AGB_{est}$.

## 4.2. Capturing the variation in tropical tree structure

We finish with a brief comment on the potential applications of the terrestrial lidar data beyond AGB estimation. The harvest measurements identified that for three out of the four trees, the majority of AGB was most often stored in the crown, with the crown mass ratio ranging from 0.42 to 0.62 (figure 3). Both this result, and the variation itself, is consistent with previously published data for large tropical trees in forests similar to our site [13], and also somewhat in agreement with data collected from across the tropics [12,46]. However here, the lidar point clouds (figure 3), which provide a unique perspective on tree structure, can begin to offer some insights into this variability. That is, these clouds permit quantification of crown form: the crown-to-tree height ratio was 0.55, 0.40, 0.54 and 0.56 for T1–T4, respectively, and the crown aspect ratio (diameter of the major axis over crown height) was 1.15, 1.63, 1.18 and 1.44, respectively. A relatively shallow, albeit wider crown then, provides a structural reason for T2 being the only sampled tree with the majority of AGB in the stem (it is perhaps also worth noting T2 was the only tree whose crown was found in the emergent layer of the stand). The underlying causal explanation for the form of a tree's crown is the result of an intricate balancing act between multiple genetic and ecosystem factors [47]. The QSMs then (figure 5), which explicitly describe three-dimensional tree architecture through to high order branching, provide a novel platform for ecologists to explore these interactions between tree architecture and ecological function [48,49].

## 5. Conclusion

In conclusion, we harvested four large tropical rainforest trees in an intact old-growth forest stand in East Amazonia, and directly weighed their above-ground green mass. We also collected detailed moisture content and woody tissue density measurements throughout each tree. We first presented some rare ecological insights provided by these data, including on the considerable structural variability of these trees, in terms of both the distribution of mass between stem and crown, and the substantial intra-tree variations in woody tissue density. We then assessed the performance of new terrestrial lidar-based methods for non-destructive estimation of the AGB of these trees, using high-quality three-dimensional measurements collected pre-harvest. This study presented, to our knowledge, the first validation of such estimates from tropical forests using reference data that were derived entirely from direct measurement. We found these estimates were accurate, and a 15-fold improvement over counterparts derived from classical allometry. The results also provided evidence to suggest error in these estimates was random and additive. If this performance were transferable across forest scenes, then these terrestrial lidar methods would enable significantly more accurate estimates of larger-scale carbon stocks. We suggested that a more robust understanding of the error in lidar-derived AGB estimates can only be achieved through the collection of further validation data.

Data accessibility. Data from the destructive measurements are archived at https://doi.org/10.5281/zenodo.4056899. Data from the terrestrial lidar measurements are archived at https://doi.org/10.5281/zenodo.4056903. TREESEG is available at https://github.com/apburt/treeseg, and the version used in this paper, v. 0.2.0, is archived at https://doi.org/10.5281/zenodo.3739213. TLSEPARATION is available at https://github.com/TLSeparation, and the version used in this paper, v. 1.2.1.5, is archived at https://doi.org/10.5281/zenodo.1147706. TREEQSM is available at https://github.com/InverseTampere/TreeQSM, and the version used in this paper, v. 2.3.2, is archived at https://doi.org/10.5281/zenodo.3560555. OPTQSM is available at https://github.com/apburt/optqsm, and the version used in this paper, v. 0.1.0, is archived at https://doi.org/10.5281/zenodo.3911269.

Authors' contributions. All authors designed the methods, and collected and analysed the data. A.B. wrote the manuscript, with contributions from all authors.

Competing interests. We declare we have no competing interests.

Funding. A.B. and M.D. acknowledge funding from Natural Environment Research Council (NERC) grant no. NE/N00373X/1 and European Research Council grant no. 757526. A.C.L.d.C. acknowledges funding from Conselho Nacional de Desenvolvimento Científico e Tecnológico (CNPq) grant no. 457914/2013-0/MCTI/CNPq/FNDCT/LBA/ESECAFLOR. P.M. acknowledges funding from NERC grant no. NE/N006852/1. L.R. acknowledges funding from NERC independent fellowship grant no. NE/N014022/1. M.D. also acknowledges funding from NERC National Centre for Earth Observation (NCEO, NE/R016518/1).

Acknowledgements. We thank ICMBio (Instituto Chico Mendes de Conservação da Biodiversidade) for providing the land to conduct this experiment. We thank the Museu Paraense Emílio Goeldi for providing logistical support. We gratefully acknowledge Cleidemar Araujo de Souza, Filomeno Martins do Amaral Filho, Josenildo Costa Amaral, Juscelino Costa Amaral, Moises Moraes Alves and Raimundo de Souza Brasão Júnior for collecting the harvest data.

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
