## [Peer Review File · Royal Society Open Science]

Review History

RSOS-201458.R0 (Original submission)

Review form: Reviewer 1 (Atticus Stovall)

Is the manuscript scientifically sound in its present form?

Yes

Are the interpretations and conclusions justified by the results?

Yes

Is the language acceptable?

Yes

Do you have any ethical concerns with this paper?

No

Have you any concerns about statistical analyses in this paper?

No

Recommendation?

Major revision is needed (please make suggestions in comments)

Comments to the Author(s)

The study details a destructive sampling and TLS validation experiment in tropical forests. By sampling 4 large tropical trees the authors show support for using TLS in this context over allometric equations. Overall, excellent work from the authors – all should be commended to taking on such an arduous task of destructive sampling in tropical forest (or any forest!) and the writing is excellent/clear/etc. Very nice work. The analysis of density was very interesting and clearly points to a major area of future research. My criticisms are not explicitly in the work presented – the science is sound and results encouraging – but I feel the authors stopped short of analyses that are well within reach of the available data. I detail these suggestions below.

First, though destructive sampling is clearly a massive undertaking, I was surprised by the simplicity of the analysis conducted given the study only included a total of 4 trees. This is with full recognition of the extreme amount of effort involved in destructive sampling. Given the low sample size I would have expected a much more detailed analysis on within-tree variation in biomass or volume. Instead the analysis focused on whole tree comparisons.

Within the TLS community there is broad agreement that whole-tree biomass/volume estimates can provide ~10% RMSE compared to destructive estimates and is much better than allometric equations. The state-of-the-art is not necessarily in whole tree biomass estimates, but in capturing within-tree variation to make sure we are not getting the “right answer for the wrong reasons.” Here is a perfect opportunity to take advantage of a major knowledge gap in TLS applications, while providing much more robust justification for the relatively small sample size in this study.

I would suggest, at the very least, evaluating the vertical distributions of volumetric estimates within trees. Even better this could be evaluated in terms of biomass with and without the measured within tree density measurements. These analyses are all possible with the described data. If the authors plan on adding this to another separate paper, it feels a bit incremental and unnecessary to separate this analysis as its own work. Given the richness and rarity in this type of destructive sampling data it would be a major contribution to the TLS community to do these analyses.

Finally, I partially question the choice of journal for this work. It may be more suited to a forest specific or biomass/carbon centric journal. Maybe *Forest Ecology and Management* or *Carbon Balance and Management* would be more within the scope – especially if the total biomass values are the focal point of the study.

Otherwise, this is nice work and warrants publication, but I see some real potential with the available data that could be used more advantageously.

Specific comments:

Line 44: There should be some clarification that this paragraph highlights TLS studies with destructive validation data using the QSM approach. Over the years there have been several other destructive TLS validation studies not described here. I would suggest adding those studies for completeness or clarify you are limiting the scope of validation studies by those using the QSM algorithm.

Review form: Reviewer 2

Is the manuscript scientifically sound in its present form?

Yes

Are the interpretations and conclusions justified by the results?

Yes

Is the language acceptable?

Yes

Do you have any ethical concerns with this paper?

Yes

Have you any concerns about statistical analyses in this paper?

No

Recommendation?

Accept with minor revision (please list in comments)

Comments to the Author(s)

The work presented in this manuscript can be divided into two parts:

- Makes relevant and rare ecological/biophysical analyses on the distribution of wood density (WD) and allocation of aboveground biomass (AGB) within different parts of the trees. It applies a well-designed and highly reliable tree harvesting (destructive) technique. Results are unique and surprising, namely the fact that the majority of the sampled AGB is stored in crowns (and not in the stems) and that WD (crucial to convert plant material volume into AGB) varies significantly with tree height and with cross sections of the tree stem.

- The referred AGB values are then used as a reference to assess the uncertainty associated with two non-destructive techniques to estimate AGB by converting tree structural metrics (e.g., stem volume, stem diameter, tree height) into AGB. 1) The commonly used allometric approach estimates AGB using existing allometric models and measurements of tree height, stem diameter along with WD. 2) The terrestrial laser scanning (TLS) technique measures tree volume that is converted into AGB using existing values of WD.

The authors found that the TLS technique surpasses the classic allometric approach with no bias associated. The authors advocate that TLS can be used to calculate plot-level AGB by summing up tree-level results with improved accuracy compared to the allometric approach. However, the TLS is acquired in an operationally unrealistic environment (the surrounding vegetation has been clear out in order to properly measure the trees individually) and it is still unclear how the TLS uncertainty would propagate in realistic environment with intrinsic obstruction effects from neighbor's vegetation.

Works in the field of estimating tree-level AGB along with the uncertainty associated are critically needed to advance our knowledge on the distribution of biomass stored in forests and hence to the global carbon cycle. Individual trees are the basic unit for estimating AGB at the plot- and landscape levels and are used by satellite remote sensing techniques for calibration and validation activities. The findings presented in the manuscript are based on only 4 tree specimens that is a limitation but worth publishing because tree harvesting AGB measurements with coincident remote sensing measurements are extremely rare.

In the following, I describe minor and major issues I have with this paper before I can recommend it for publication. I hope that the authors make use of it to improve the manuscript:

Major:

1- The text is often long and confusing. Language and techniques descriptions could be simplified to reach a broader public:

1.1. For instance, the authors decided to use jargon that is often long and difficult to digest or follow (e.g. “woody tissue density” instead of commonly used “wood density”; “whole-tree green mass” instead of “leaves mass?” etc.)

1.2. Technical concepts should be better explained. For example, is not clear from the very beginning how the lidar technique estimates AGB. It calculates tree volume and converts it to AGB using reference wood density values? Or it calculates tree structural metrics (tree height, stem diameter) and uses the allometric equations to find AGB? Therefore, I had difficulties in identifying the difference between the lidar and allometric approach.

2- The abstract is not clear, it is hard to identify the main goals of the paper. I believe the reasoning could be improved. I like the idea of dividing the work into two steps (as described in the introduction of this review) but I’m not sure whether it is the best approach.

3- The lidar approach is here called non-destructive (just to make clear, I agree with the notation) but there was a lot of destruction in order to clear surrounding vegetation and open path to the laser beams. As you state in the “Discussion” section, it is still unclear the ability of the TLS to measure those trees in a natural environment and how would that propagate to the final results. Results wouldn’t be that impressive.

3.1. Did you consider doing TLS measurements before and after the clearing surrounding vegetation to assess uncertainty associated with the obstruction of the signal in natural environments?

3.2. Is there any literature on 3.1?

3.3. Is there any “ethical” guidelines for harvesting trees for scientific purposes? Were these trees condemned to be harvested? You mention that they are located in private land.

4- How do you derive DBH and tree height to feed Chave’s model (Eq.1)? Are these derived from the TLS? It is not clear.

5- I might have missed this point. Regarding the TLS approach, did you provide estimates on the uncertainty associated with using “averaged” WD as opposed to the detailed and varying WD measurements?

5.1. Would impact both approaches (TLS and allometric) in a similar magnitude?

5.2. Are errors of the order of >20% as found by Momo Takoudjou et al and Gonzales de Tango et al. ?

6- I think there is a discussion missing on the topic of tree-level AGB estimates and the extrapolation to ecological meaningful scales such as the landscape- to regional-level. TLS has a great potential to derive local reference AGB and refine tree allometry (e.g. DBH-crown size, DBH-tree height). However, similarly to traditional field techniques, TLS has spatial and temporal coverage limitations.

6.1 - Could you discuss the synergies between tree-level measurements from TLS and airborne lidar scanning (ALS)? There are now techniques to massively measure tree height and crown size from ALS in the tropics (e.g. Ferraz et al 2020, 2016) that can be converted into DBH estimates

using pantropical or local allometry that converts crown size into DBH Jucker et al 2017 or Figueiredo et al 2016.

6.2. What is the role of TLS to locally define or refine tree allometry on crown size to DBH or AGB that could be then used by ALS crown measurements to calculate AGB at large scale?

6.2 The fact that most of the AGB is allocated to the crown should be emphasized and it supports the fact that ALS crown measurements can have a larger role in calculating AGB baselines as well as that allometric equations that include crown size should be further developed. This is an important point for TLS or for merging TLS and ALS.

6.3. Did you compare your results with similar work on the AGB stored in the crowns from Goodman et al 2014 (30%?) ? I believe there is more work on this.

Minor:

Line 25-26: It is not clear that "lidar AGB" is calculated from converting measured tree volume along with wood density.

Line 37: I believe (intact) old-growth forest is commonly used in tropical ecology and science.

Line 22: What volume? Total tree green mass volume? Volume of the crown envelope, stem volume? Not clear.

Line 43-47: This is amazing, it gives us an instantaneous perception of the uncertainties associated with landscape, regional and global estimates. I would state it in the abstract.

Line 73-80: I agree, but TLS has high spatial limitations. What is your opinion on estimates from tree crown measurements using airborne lidar (e.g. (Ferraz et al 2020, 2016)) that can be then converted into biomass using regional or pantropical crown-ABG allometry as proposed in (Jucker et al 2017) or (Figueiredo et al 2016) in particular for the Brazilian Amazon?

Line 73-80: Also, what is your opinion on the synergies between TLS and ALS. I guess ALS has improved ability to measure crown size (at least diameter/perimeter) of large trees compared to TLS. Then, crowns might store the majority of the biomass as you state in the abstract.

Line 94: question #1 and #2 are basically the same question and highly correlated to #3. Summarize?

Line 103-104: This might be out of the scope of your work, but do you think that the lack of relationship between error and scale might be due to competition factors (open forest) or a singularity of the *Eucalyptus* spp. Species? This is intriguing.

Line 213:135: The fact of studying only 4 trees is not a limitation of the work (it's time and resources expensive work) and I would say it earlier in the text, even in the abstract.

Line 138-139: I think these are quite important and relevant analyses and findings. Good you emphasize it in the abstract.

Line 151: old-growth and mixed species is redundant, I believe. Is it intact forest? Somewhere in the text you call it natural but it is not clear what natural means. Is it naturally intact, natural regrowth?

Line 159: I never heard about “felling area” in tropical ecology and management. What does that mean?

Line 166: 3 trees are buttress trees. Do you think that might impact the result? they might support larger crowns, taller trees and higher ABG compared with no-buttress trees?

Line 164-line174: I agree with the notation non-destructive methods but in fact there is a significant clearing for proper measurements. It means that in a practical situation the TLS measurements would be impacted by surrounding vegetation. What would be the errors associated with that? Is there any literature on that?

Line 181: tarpaulin? Excellent, good to know!

Line 178-203: remarkable work.

Line 247-248: The TLS and the software does not estimate biomass, they estimate tree metrics that are used to estimate biomass using auxiliary WD.

Figure 3. I would suggest to combine both panels by including the bars in the upper panel. This are surprising results, Goodman found about 30% only if I can recall correctly.

Figure 4. Impressive results, namely the intra-variability and the bias. I don't understand the calculation of the solid grey line by interpolation, though.

Figure 5. Does this image has added value compared to Figure 3?

Line 353-355. Was the error correlated with crown size or DBH? Also, is the crown size (e.g., diameter) closely related with dbh compared to with tree height?

Lines 356-361. Exactly, means that the estimates are not biased, crucial when scaling up to landscape or regional-level. This is quite important although only based on 4 trees.

Lines 395-line 399. I guess you could account for some errors, such as for example the occlusion effect. For that you could have measure the trees before and after removing the surrounding vegetation. Did you consider that?

Ferraz A, Saatchi S, Longo M and Clark D 2020 Tropical tree size–frequency distributions from airborne lidar *Ecol. Appl.* 0

Ferraz A, Saatchi S, Mallet C and Meyer V 2016 Lidar detection of individual tree size in tropical forests *Remote Sens. Environ.* 183 318–33 Online: <http://dx.doi.org/10.1016/j.rse.2016.05.028>

Figueiredo E, D'Oliveira M, Braz E, de Almeida Papa D and Fearnside P M 2016 LIDAR-based estimation of bole biomass for precision management of an Amazonian forest: Comparisons of ground-based and remotely sensed estimates *Remote Sens. Environ.* 187 281–93 Online: <http://dx.doi.org/10.1016/j.rse.2016.10.026>

Goodman R, Phillips O and Baker T 2014 The importance of crown dimensions to improve tropical tree biomass estimates *Ecol. Appl.* 24 680–98 Online: <http://dx.doi.org/10.1890/13-0070.1>

Jucker T, Caspersen J, Chave J, Antin C, Barbier N, Bongers F, Dalponte M, van Ewijk K Y, Forrester D I, Haeni M, Higgins S I, Holdaway R J, Iida Y, Lorimer C, Marshall P L, Momo S, Moncrieff G R, Ploton P, Poorter L, Rahman K A, Schlund M, Sonké B, Sterck F J, Trugman A T, Usoltsev V A, Vanderwel M C, Waldner P, Wedeux B M M, Wirth C, Wöll H, Woods M, Xiang W,

Zimmermann N E and Coomes D A 2017 Allometric equations for integrating remote sensing imagery into forest monitoring programmes Glob. Chang. Biol. 23 177–90

Decision letter (RSOS-201458.R0)

Dear Dr Burt

On behalf of the Editors, we are pleased to inform you that your Manuscript RSOS-201458 "New insights into large tropical tree mass and structure from direct harvest and terrestrial lidar" has been accepted for publication in Royal Society Open Science subject to minor revision in accordance with the referees' reports. Please find the referees' comments along with any feedback from the Editors below my signature.

Please submit your revised manuscript and required files (see below) no later than 7 days from today's (ie 23-Nov-2020) date. Note: the ScholarOne system will 'lock' if submission of the revision is attempted 7 or more days after the deadline. If you do not think you will be able to meet this deadline please contact the editorial office immediately.

on behalf of Dr Yhasmin Mendes de Moura (Associate Editor) and Pete Smith (Subject Editor)
openscience@royalsociety.org

Associate Editor Comments to Author (Dr Yhasmin Mendes de Moura):
Associate Editor: 1
Comments to the Author:
Dear authors,

I particularly enjoy reading your paper, and strong believe it will make great contributions for the TLS community, especially for the destructive experiments conducted. Congratulations for the effort!

However, the reviewers brought up some important points that should be read and implemented by the authors. I recommend this paper to be accepted with minor revisions after implementing reviewers comments.

Best regards,

Reviewer comments to Author:

Reviewer: 1

Comments to the Author(s)

The study details a destructive sampling and TLS validation experiment in tropical forests. By sampling 4 large tropical trees the authors show support for using TLS in this context over allometric equations. Overall, excellent work from the authors – all should be commended to taking on such an arduous task of destructive sampling in tropical forest (or any forest!) and the writing is excellent/clear/etc. Very nice work. The analysis of density was very interesting and clearly points to a major area of future research. My criticisms are not explicitly in the work presented – the science is sound and results encouraging – but I feel the authors stopped short of analyses that are well within reach of the available data. I detail these suggestions below.

First, though destructive sampling is clearly a massive undertaking, I was surprised by the simplicity of the analysis conducted given the study only included a total of 4 trees. This is with full recognition of the extreme amount of effort involved in destructive sampling. Given the low sample size I would have expected a much more detailed analysis on within-tree variation in biomass or volume. Instead the analysis focused on whole tree comparisons.

Within the TLS community there is broad agreement that whole-tree biomass/volume estimates can provide ~10% RMSE compared to destructive estimates and is much better than allometric equations. The state-of-the-art is not necessarily in whole tree biomass estimates, but in capturing within-tree variation to make sure we are not getting the “right answer for the wrong reasons.” Here is a perfect opportunity to take advantage of a major knowledge gap in TLS applications, while providing much more robust justification for the relatively small sample size in this study.

I would suggest, at the very least, evaluating the vertical distributions of volumetric estimates within trees. Even better this could be evaluated in terms of biomass with and without the measured within tree density measurements. These analyses are all possible with the described data. If the authors plan on adding this to another separate paper, it feels a bit incremental and unnecessary to separate this analysis as its own work. Given the richness and rarity in this type of destructive sampling data it would be a major contribution to the TLS community to do these analyses.

Finally, I partially question the choice of journal for this work. It may be more suited to a forest specific or biomass/carbon centric journal. Maybe Forest Ecology and Management or Carbon Balance and Management would be more within the scope – especially if the total biomass values are the focal point of the study.

Otherwise, this is nice work and warrants publication, but I see some real potential with the available data that could be used more advantageously.

Specific comments:

Line 44: There should be some clarification that this paragraph highlights TLS studies with destructive validation data using the QSM approach. Over the years there have been several other destructive TLS validation studies not described here. I would suggest adding those studies for completeness or clarify you are limiting the scope of validation studies by those using the QSM algorithm.

Reviewer: 2

Comments to the Author(s)

The work presented in this manuscript can be divided into two parts:

- Makes relevant and rare ecological/biophysical analyses on the distribution of wood density (WD) and allocation of aboveground biomass (AGB) within different parts of the trees. It applies a well-designed and highly reliable tree harvesting (destructive) technique. Results are unique and surprising, namely the fact that the majority of the sampled AGB is stored in crowns (and not in the stems) and that WD (crucial to convert plant material volume into AGB) varies significantly with tree height and with cross sections of the tree stem.

- The referred AGB values are then used as a reference to assess the uncertainty associated with two non-destructive techniques to estimate AGB by converting tree structural metrics (e.g., stem volume, stem diameter, tree height) into AGB. 1) The commonly used allometric approach estimates AGB using existing allometric models and measurements of tree height, stem diameter along with WD. 2) The terrestrial laser scanning (TLS) technique measures tree volume that is converted into AGB using existing values of WD.

The authors found that the TLS technique surpasses the classic allometric approach with no bias associated. The authors advocate that TLS can be used to calculate plot-level AGB by summing up tree-level results with improved accuracy compared to the allometric approach. However, the TLS is acquired in an operationally unrealistic environment (the surrounding vegetation has been clear out in order to properly measure the trees individually) and it is still unclear how the TLS uncertainty would propagate in realistic environment with intrinsic obstruction effects from neighbor's vegetation.

Works in the field of estimating tree-level AGB along with the uncertainty associated are critically needed to advance our knowledge on the distribution of biomass stored in forests and hence to the global carbon cycle. Individual trees are the basic unit for estimating AGB at the plot- and landscape levels and are used by satellite remote sensing techniques for calibration and validation activities. The findings presented in the manuscript are based on only 4 tree specimens that is a limitation but worth publishing because tree harvesting AGB measurements with coincident remote sensing measurements are extremely rare.

In the following, I describe minor and major issues I have with this paper before I can recommend it for publication. I hope that the authors make use of it to improve the manuscript:

Major:

1- The text is often long and confusing. Language and techniques descriptions could be simplified to reach a broader public:

1.1. For instance, the authors decided to use jargon that is often long and difficult to digest or follow (e.g. "woody tissue density" instead of commonly used "wood density"; "whole-tree green mass" instead of "leaves mass?" etc.)

1.2. Technical concepts should be better explained. For example, is not clear from the very beginning how the lidar technique estimates AGB. It calculates tree volume and converts it to AGB using reference wood density values? Or it calculates tree structural metrics (tree height, stem diameter) and uses the allometric equations to find AGB? Therefore, I had difficulties in identifying the difference between the lidar and allometric approach.

2- The abstract is not clear, it is hard to identify the main goals of the paper. I believe the reasoning could be improved. I like the idea of dividing the work into two steps (as described in the introduction of this review) but I'm not sure whether it is the best approach.

3- The lidar approach is here called non-destructive (just to make clear, I agree with the notation) but there was a lot of destruction in order to clear surrounding vegetation and open path to the laser beams. As you state in the "Discussion" section, it is still unclear the ability of the TLS to measure those trees in a natural environment and how would that propagate to the final results. Results wouldn't be that impressive.

3.1. Did you consider doing TLS measurements before and after the clearing surrounding vegetation to assess uncertainty associated with the obstruction of the signal in natural environments?

3.2. Is there any literature on 3.1?

3.3. Is there any "ethical" guidelines for harvesting trees for scientific purposes? Were these trees condemned to be harvested? You mention that they are located in private land.

4- How do you derive DBH and tree height to feed Chave's model (Eq.1)? Are these derived from the TLS? It is not clear.

5- I might have missed this point. Regarding the TLS approach, did you provide estimates on the uncertainty associated with using "averaged" WD as opposed to the detailed and varying WD measurements?

5.1. Would impact both approaches (TLS and allometric) in a similar magnitude?

5.2. Are errors of the order of >20% as found by Momo Takoudjou et al and Gonzales de Tango et al. ?

6- I think there is a discussion missing on the topic of tree-level AGB estimates and the extrapolation to ecological meaningful scales such as the landscape- to regional-level. TLS has a great potential to derive local reference AGB and refine tree allometry (e.g. DBH-crown size, DBH-tree height). However, similarly to traditional field techniques, TLS has spatial and temporal coverage limitations.

6.1 - Could you discuss the synergies between tree-level measurements from TLS and airborne lidar scanning (ALS)? There are now techniques to massively measure tree height and crown size from ALS in the tropics (e.g. Ferraz et al 2020, 2016) that can be converted into DBH estimates using pantropical or local allometry that converts crown size into DBH Jucker et al 2017 or Figueiredo et al 2016.

6.2. What is the role of TLS to locally define or refine tree allometry on crown size to DBH or AGB that could be then used by ALS crown measurements to calculate AGB at large scale?

6.2 The fact that most of the AGB is allocated to the crown should be emphasized and it supports the fact that ALS crown measurements can have a larger role in calculating AGB baselines as well as that allometric equations that include crown size should be further developed. This is an important point for TLS or for merging TLS and ALS.

6.3. Did you compare your results with similar work on the AGB stored in the crowns from Goodman et al 2014 (30%)? I believe there is more work on this.

Minor:

Line 25-26: It is not clear that "lidar AGB" is calculated from converting measured tree volume along with wood density.

Line 37: I believe (intact) old-growth forest is commonly used in tropical ecology and science.

Line 22: What volume? Total tree green mass volume? Volume of the crown envelope, stem volume? Not clear.

Line 43-47: This is amazing, it gives us an instantaneous perception of the uncertainties associated with landscape, regional and global estimates. I would state it in the abstract.

Line 73-80: I agree, but TLS has high spatial limitations. What is your opinion on estimates from tree crown measurements using airborne lidar (e.g. (Ferraz et al 2020, 2016)) that can be then converted into biomass using regional or pantropical crown-ABG allometry as proposed in (Jucker et al 2017) or (Figueiredo et al 2016) in particular for the Brazilian Amazon?

Line 73-80: Also, what is your opinion on the synergies between TLS and ALS. I guess ALS has improved ability to measure crown size (at least diameter/perimeter) of large trees compared to TLS. Then, crowns might store the majority of the biomass as you state in the abstract.

Line 94: question #1 and #2 are basically the same question and highly correlated to #3. Summarize?

Line 103-104: This might be out of the scope of your work, but do you think that the lack of relationship between error and scale might be due to competition factors (open forest) or a singularity of the *Eucalyptus* spp. Species? This is intriguing.

Line 213:135: The fact of studying only 4 trees is not a limitation of the work (it's time and resources expensive work) and I would say it earlier in the text, even in the abstract.

Line 138-139: I think these are quite important and relevant analyses and findings. Good you emphasize it in the abstract.

Line 151: old-growth and mixed species is redundant, I believe. Is it intact forest? Somewhere in the text you call it natural but it is not clear what natural means. Is it naturally intact, natural regrowth?

Line 159: I never heard about "felling area" in tropical ecology and management. What does that mean?

Line 166: 3 threes are buttress trees. Do you think that might impact the result? they might support larger crowns, taller trees and higher ABG compared with no-buttress trees?

Line 164-line174: I agree with the notation non-destructive methods but in fact there is a significant clearing for proper measurements. It's means than a practical situation the TLS measurements would be impacted by surrounding vegetation. What would be the errors associated with that? Is there any literature on that?

Line 181: tarpaulin? Excellent, good to know!

Line 178-203: remarkable work.

Line 247-248: The TLS and the software does not estimate biomass, they estimate tree metrics that are used to estimate biomass using auxiliary WD.

Figure 3. I would suggest to combine both panels by including the bars in the upper panel. This are surprising results, Goodman found about 30% only if I can recall correctly.

Figure 4. Impressive results, namely the intra-variability and the bias. I don't understand the calculation of the solid grey line by interpolation, though.

Figure 5. Does this image has added value compared to Figure 3?

Line 353-355. Was the error correlated with crown size or DBH? Also, is the crown size (e.g., diameter) closely related with dbh compared to with tree height?

Lines 356-361. Exactly, means that the estimates are not biased, crucial when scaling up to landscape or regional-level. This is quite important although only based on 4 trees.

Lines 395-line 399. I guess you could account for some errors, such as for example the occlusion effect. For that you could have measure the trees before and after removing the surrounding vegetation. Did you consider that?

Ferraz A, Saatchi S, Longo M and Clark D 2020 Tropical tree size-frequency distributions from airborne lidar *Ecol. Appl.* 0

Ferraz A, Saatchi S, Mallet C and Meyer V 2016 Lidar detection of individual tree size in tropical forests *Remote Sens. Environ.* 183 318–33 Online: <http://dx.doi.org/10.1016/j.rse.2016.05.028>

Figueiredo E, D'Oliveira M, Braz E, de Almeida Papa D and Fearnside P M 2016 LIDAR-based estimation of bole biomass for precision management of an Amazonian forest: Comparisons of ground-based and remotely sensed estimates *Remote Sens. Environ.* 187 281–93 Online: <http://dx.doi.org/10.1016/j.rse.2016.10.026>

Goodman R, Phillips O and Baker T 2014 The importance of crown dimensions to improve tropical tree biomass estimates *Ecol. Appl.* 24 680–98 Online: <http://dx.doi.org/10.1890/13-0070.1>

Jucker T, Caspersen J, Chave J, Antin C, Barbier N, Bongers F, Dalponte M, van Ewijk K Y, Forrester D I, Haeni M, Higgins S I, Holdaway R J, Iida Y, Lorimer C, Marshall P L, Momo S, Moncrieff G R, Ploton P, Poorter L, Rahman K A, Schlund M, Sonké B, Sterck F J, Trugman A T, Usoltsev V A, Vanderwel M C, Waldner P, Wedeux B M M, Wirth C, Wöll H, Woods M, Xiang W, Zimmermann N E and Coomes D A 2017 Allometric equations for integrating remote sensing imagery into forest monitoring programmes *Glob. Chang. Biol.* 23 177–90

===PREPARING YOUR MANUSCRIPT===

===PREPARING YOUR REVISION IN SCHOLARONE===

- An individual file of each figure (EPS or print-quality PDF preferred [either format should be produced directly from original creation package], or original software format).
 - An editable file of each table (.doc, .docx, .xls, .xlsx, or .csv).
 - An editable file of all figure and table captions.
- Note: you may upload the figure, table, and caption files in a single Zip folder.
- Any electronic supplementary material (ESM).
 - If you are requesting a discretionary waiver for the article processing charge, the waiver form must be included at this step.
 - If you are providing image files for potential cover images, please upload these at this step, and inform the editorial office you have done so. You must hold the copyright to any image provided.
 - A copy of your point-by-point response to referees and Editors. This will expedite the preparation of your proof.

- Ensure that your data access statement meets the requirements at <https://royalsociety.org/journals/authors/author-guidelines/#data>. You should ensure that you cite the dataset in your reference list. If you have deposited data etc in the Dryad repository, please only include the 'For publication' link at this stage. You should remove the 'For review' link.
- If you are requesting an article processing charge waiver, you must select the relevant waiver option (if requesting a discretionary waiver, the form should have been uploaded at Step 3 'File upload' above).
- If you have uploaded ESM files, please ensure you follow the guidance at <https://royalsociety.org/journals/authors/author-guidelines/#supplementary-material> to include a suitable title and informative caption. An example of appropriate titling and captioning may be found at https://figshare.com/articles/Table_S2_from_Is_there_a_trade-off_between_peak_performance_and_performance_breadth_across_temperatures_for_aerobic_scops_in_teleost_fishes_/3843624.

Author's Response to Decision Letter for (RSOS-201458.R0)

See Appendix A.

Decision letter (RSOS-201458.R1)

Dear Dr Burt,

It is a pleasure to accept your manuscript entitled "New insights into large tropical tree mass and structure from direct harvest and terrestrial lidar" in its current form for publication in Royal Society Open Science. The comments of the reviewer(s) who reviewed your manuscript are included at the foot of this letter.

Please can you provide an amended 'author contribution' statement as soon as possible - the current iteration is not sufficiently detailed. Per our instructions to authors, we ask that individual contributions are delineated for each author, and it is not acceptable to say that "All authors contributed".

on behalf of Dr Yhasmin Mendes de Moura (Associate Editor) and Pete Smith (Subject Editor)
openscience@royalsociety.org

Associate Editor Comments to Author (Dr Yhasmin Mendes de Moura):
Associate Editor
Comments to the Author:
Dear authors,

After carefully reading your response to the reviewers I believe you had addressed the major concerns. Therefore, I would recommend your manuscript for publication.

Best regards,
Follow Royal Society Publishing on Twitter: @RSocPublishing
Follow Royal Society Publishing on Facebook:
<https://www.facebook.com/RoyalSocietyPublishing.FanPage/>
Read Royal Society Publishing's blog:
<https://royalsociety.org/blog/blogsearchpage/?category=Publishing>

Appendix A

Dear editors,

We sincerely thank both reviewers for providing critical assessments of our manuscript. Please find attached our revised manuscript.

Overall, we have incorporated the large majority of the comments into the paper. This has led to various changes, most significantly: i) in response to reviewer 1, some additional results. This includes a new figure (figure 6) and some additional text in the results and discussion sections. ii) In response to reviewer 2, we have rewritten the abstract, and revised three figures (figures 3, 5 and 7).

Below is our response to each of the reviewers' comments.

Regards,

The authors

REVIEWER 1

The study details a destructive sampling and TLS validation experiment in tropical forests. By sampling 4 large tropical trees the authors show support for using TLS in this context over allometric equations. Overall, excellent work from the authors – all should be commended to taking on such an arduous task of destructive sampling in tropical forest (or any forest!) and the writing is excellent/clear/etc. Very nice work. The analysis of density was very interesting and clearly points to a major area of future research. My criticisms are not explicitly in the work presented – the science is sound and results encouraging – but I feel the authors stopped short of analyses that are well within reach of the available data. I detail these suggestions below.

We thank reviewer 1 for taking the time to review the manuscript, and for providing suggestions on how to improve the paper. Below, for each set of comments, we describe the changes that have been made.

First, though destructive sampling is clearly a massive undertaking, I was surprised by the simplicity of the analysis conducted given the study only included a total of 4 trees. This is with full recognition of the extreme amount of effort involved in destructive sampling. Given the low sample size I would have expected a much more detailed analysis on within-tree variation in biomass or volume. Instead the analysis focused on whole tree comparisons.

Within the TLS community there is broad agreement that whole-tree biomass/volume estimates can provide ~10% RMSE compared to destructive estimates and is much better than allometric equations. The state-of-the-art is not necessarily in whole tree biomass estimates, but in capturing within-tree variation to make sure we are not getting the “right answer for the wrong reasons.” Here is a perfect opportunity to take advantage of a major knowledge gap in TLS applications, while providing much more robust justification for the relatively small sample size in this study.

I would suggest, at the very least, evaluating the vertical distributions of volumetric estimates within trees. Even better this could be evaluated in terms of biomass with and without the measured within tree density measurements. These analyses are all possible with the described data. If the authors plan on adding this to another separate paper, it feels a bit incremental and unnecessary to separate this analysis as its own work. Given the richness and rarity in this type of destructive sampling data it would be a major contribution to the TLS community to do these analyses.

We agree with reviewer 1 that these additional analyses (i.e., exploring the accuracy of lidar-derived estimates of AGB at a resolution finer than whole-tree AGB) would substantially add to the manuscript. Unfortunately, our particular harvesting methods permit only superficial exploration of this. That is, here, directly measured green mass was allocated to just two pools (stem and crown), rather than at a more low-level (e.g., by height bins or branching order). This prevents us from validating vertical distributions of AGB generated from the lidar estimation methods.

But this does allow us to explore the accuracy with which the lidar methods estimate the AGB stored in the stem and crown of each tree. This analysis has now been undertaken, and is included in the revised manuscript via a new figure (figure 6): this figure compares the distributions of AGB between the stem and crown of each harvested tree, for both the reference measurements and lidar estimates.

The following paragraph has been included in the results section (line 339): ‘The lidar methods also accurately estimated the allocation of AGB intra-tree, as illustrated in figure 6. This compares the distributions of AGB between the stem and crown, for both these estimates and the reference measurements. The distributions are similar: 48%, 58%, 38% and 41% of harvest-derived AGB_{ref} was found in the stems of T1-T4 respectively, whilst 48%, 62%, 42% and 47% of lidar-derived AGB_{est} was respectively allocated to each stem.’

The following two sentences have been included in the discussion section (line 431): ‘This is further corroborated by figure 6, which demonstrates that the lidar methods were able to accurately resolve the allocation of AGB between the stem and crown of each tree. That is, AGB_{est} did not accidentally agree with AGB_{ref} because of underestimation in the stem and overestimation in the crown, but because of close agreement in both pools.’

Finally, I partially question the choice of journal for this work. It may be more suited to a forest specific or biomass/carbon centric journal. Maybe Forest Ecology and Management or Carbon Balance and Management would be more within the scope – especially if the total biomass values are the focal point of the study.

We felt there were two distinct features of this work. First, the rare ecological insights provided by the harvest data (e.g., the intra-tree density variations). Second, the comparison between non-destructive approaches to estimating AGB. We therefore thought this paper may be of interest beyond readers of carbon-centric journals, and that a general scientific journal, such as Royal Society Open Science, would be an appropriate choice.

Reviewer 2 has suggested a rewrite of the abstract; in particular that the abstract should more clearly divide the work into these two components. We have incorporated this into the updated manuscript, and hope this will now help the paper appeal to a wider readership.

Otherwise, this is nice work and warrants publication, but I see some real potential with the available data that could be used more advantageously.

Specific comments:

Line 44: There should be some clarification that this paragraph highlights TLS studies with destructive validation data using the QSM approach. Over the years there have been several other destructive TLS validation studies not described here. I would suggest adding those studies for completeness or clarify you are limiting the scope of validation studies by those using the QSM algorithm.

This point was not clear in the original manuscript. That is, a simpler alternative approach to estimating AGB from terrestrial lidar data, is to retrieve variables of tree structure from the point clouds (e.g., stem diameter and tree height), and subsequently feed these parameters into allometric models. Given that our study does not consider this approach, we have clarified that our literature review is restricted to previous studies that validated lidar-derived estimates of AGB which were generated using above-ground woody volume estimates. The first sentence of this paragraph has been updated to read (line 115): 'To date, several publications present studies validating AGB_{est} retrieved from terrestrial lidar data via QSM-derived volume estimates.'

REVIEWER 2

The work presented in this manuscript can be divided into two parts:

- Makes relevant and rare ecological/biophysical analyses on the distribution of wood density (WD) and allocation of aboveground biomass (AGB) within different parts of the trees. It applies a well-designed and highly reliable tree harvesting (destructive) technique. Results are unique and surprising, namely the fact that the majority of the sampled AGB is stored in crowns (and not in the stems) and that WD (crucial to convert plant material volume into AGB) varies significantly with tree height and with cross sections of the tree stem.

- The referred AGB values are then used as a reference to assess the uncertainty associated with two non-destructive techniques to estimate AGB by converting tree structural metrics (e.g., stem volume, stem diameter, tree height) into AGB. 1) The commonly used allometric approach estimates AGB using existing allometric models and measurements of tree height, stem diameter along with WD. 2) The terrestrial laser scanning (TLS) technique measures tree volume that is converted into AGB using existing values of WD. The authors found that the TLS technique surpasses the classic allometric approach with no bias associated. The authors advocate that TLS can be used to calculate plot-level AGB by summing up tree-level results with improved accuracy compared to the allometric approach. However, the TLS is acquired in an operationally unrealistic environment (the surrounding vegetation has been clear out in order to properly measure the trees individually) and it is still unclear how the TLS uncertainty would propagate in realistic environment with intrinsic obstruction effects from neighbor's vegetation.

Works in the field of estimating tree-level AGB along with the uncertainty associated are critically needed to advance our knowledge on the distribution of biomass stored in forests and hence to the global carbon cycle. Individual trees are the basic unit for estimating AGB at the plot- and landscape levels and are used by satellite remote sensing techniques for calibration and validation activities. The findings presented in the manuscript are based on only 4 tree specimens that is a limitation but worth publishing because tree harvesting AGB measurements with coincident remote sensing measurements are extremely rare.

We thank reviewer 2 for taking the time to review the manuscript, and for providing suggestions on how to improve the manuscript. Below, for each set of comments, we describe the changes that have been made, or justify why we have chosen not to make the suggested changes.

In the following, I describe minor and major issues I have with this paper before I can recommend it for publication. I hope that the authors make use of it to improve the manuscript:

Major:

1- The text is often long and confusing. Language and techniques descriptions could be simplified to reach a broader public:

1.1. For instance, the authors decided to use jargon that is often long and difficult to digest or follow (e.g. “woody tissue density” instead of commonly used “wood density”; “whole-tree green mass” instead of “leaves mass?” etc.)

1.2. Technical concepts should be better explained. For example, is not clear from the very beginning how the lidar technique estimates AGB. It calculates tree volume and converts it to AGB using reference wood density values? Or it calculates tree structural metrics (tree height, stem diameter) and uses the allometric equations to find AGB? Therefore, I had difficulties in identifying the difference between the lidar and allometric approach.

We have carefully considered these comments by reviewer 2 that the manuscript is often long and confusing, contains jargon, and that this is a concern for wider readership. With respect to jargon, we have decided not to remove technical terms because each was selected to convey a specific meaning, and that these terms are not interchangeable with more commonly-used terms (discussed in the following two paragraphs). We recognise the advantages of commonly-used terms but we hope the reviewer also agrees that we need to balance this with the need for precision in terminology, especially given that we are using a relatively new technique.

Wood density vs. woody tissue density: in a tropical forest ecology / research context, we believe there is a broadly consistent acceptance of what constitutes a wood density measurement [1]. That is, widely used increment borer-based protocol (e.g., [2]) suggest samples should be taken from the stem approximately 1m above-ground, after bark has been removed. Therefore, the retrieved core will comprise of stem xylem tissue (plus possibly pith depending on depth). However, as described in the introduction (line 92), to convert a lidar-derived above-ground green woody volume into an unbiased estimate of AGB, a specific value of density is required. Amongst the other described properties, this value must reflect the density of each tissue occupying the volume (including bark), weighted by the relative quantity of each tissue. We therefore think it would be inaccurate to describe this value using the standard term ‘wood density’. There is no precedent in the literature for describing the value we require, so we carefully arrived at the term ‘above-ground basic woody tissue density’. We agree that this might seem unwieldy, but the importance of arriving at a precise, unambiguous definition outweighs this.

Green mass: this is used for woody tissue, not leaves, and this refers to a mass state where it is assumed cell walls are saturated and lumina filled with water (line 96). We would be happy to use the perhaps more intuitive term ‘wet’, but it does seem that ‘green’ is the prevailing term in the literature (e.g., [1] and [2]).

With respect to the introduction of technical concepts, here we have assumed this is directed to the abstract; i.e., the lack of an introduction to the lidar-based methods. As suggested by reviewer 2 in the next set of comments, we have now rewritten the abstract. This includes a sentence to state that the lidar-derived AGB estimates are based on volume estimates, which are themselves retrieved from the quantitative structural models (cylinder models).

[1] Williamson and Weimann (2010); ‘Measuring wood specific gravity ... Correctly’.

[2] Chave (2005); ‘Measuring wood density for tropical forest trees: a field manual’.

2- The abstract is not clear, it is hard to identify the main goals of the paper. I believe the reasoning could be improved. I like the idea of dividing the work into two steps (as described in the introduction of this review) but I’m not sure whether it is the best approach.

We have rewritten the abstract. As suggested, we have more clearly divided the work into its two distinct components; first the ecological insights provided by the harvest data, and second, the non-destructive above-ground biomass estimates. Further into this review, reviewer 2 also

suggests specific additions to the abstract, which we have incorporated. This includes two additional sentences on context: i) on previous destructive harvest measurements undertaken in the Amazon, and ii) on the lidar-derived approach to estimating AGB.

3- The lidar approach is here called non-destructive (just to make clear, I agree with the notation) but there was a lot of destruction in order to clear surrounding vegetation and open path to the laser beams. As you state in the "Discussion" section, it is still unclear the ability of the TLS to measure those trees in a natural environment and how would that propagate to the final results. Results wouldn't be that impressive.

3.1. Did you consider doing TLS measurements before and after the clearing surrounding vegetation to assess uncertainty associated with the obstruction of the signal in natural environments?

3.2. Is there any literature on 3.1?

3.3. Is there any "ethical" guidelines for harvesting trees for scientific purposes? Were these trees condemned to be harvested? You mention that they are located in private land.

We agree with reviewer 2 that this is both a fundamental point, and a limitation of the study. That is, the terrestrial lidar data were collected after neighbouring vegetation surrounding the trees of interest was cleared. This therefore raises questions about the transferability of our results. The absence of lidar measurements prior to clearing was the combination of an oversight and time constraints. To capture truly 'natural' lidar data, the scanning protocol would have had to closely mirror those employed in a non-destructive setting (e.g., scans acquired on a 10/20m uniform grid across a 1ha forest stand). Given that the harvested trees were several hundreds of metres apart, this would not have been feasible within the time available.

We are not aware of any literature exploring the impacts of data acquisition protocol on the accuracy of lidar-derived AGB estimates, but we did give some thought to potential methods to artificially degrade these data. For example, during the same campaign we also collected terrestrial lidar data from two long-term monitoring plots (approximately 7km from the harvest site). One idea was to generate a quantitative description of the occlusion in these data (i.e., the divergence of these point clouds from the theoretical expectation), and use this as a mask to degrade the harvest lidar data. Ultimately, we decided that accounting for the impacts of occlusion is a difficult problem, and such ideas are studies in their own right, and therefore outside the scope of this paper.

However, we believe our results are still of broad interest, because the high-quality data collected here provide the ability understand the minimum error that can be expected in lidar-derived AGB estimates. Overall, we think this was explained in the introduction, methods and discussion sections of the original manuscript, so we have not made any major changes.

We have updated the final sentence in the abstract to read (line 28): 'Were these results transferable across forest scenes, terrestrial lidar methods ...'

We have also added the following sentence in the introduction (line 141): 'This approach provides the opportunity to generate a robust understanding of the minimum error that can be expected in lidar-derived AGB estimates, with currently available hardware and data processing methods.'

We have also revised a sentence in the discussion (line 386): 'If these characteristics, derived from high-quality data collected on a small sample of four trees, were transferable across forest scenes ...'

The trees were harvested after obtaining appropriate permissions from the Chico Mendes Institute for Biodiversity Conservation (ICMBio) which is under the jurisdiction of the Ministry of the Environment.

4- How do you derive DBH and tree height to feed Chave's model (Eq.1)? Are these derived from the TLS? It is not clear.

Stem diameter was measured pre-harvest using a circumference/diameter tape (line 182). Tree height was measured post-harvest using a surveyor's tape measure (line 199). This is now re-stated in section 2.5.4 (line 284): 'Where here, stem diameter was measured pre-harvest using a circumference/diameter tape at 1.3m above-ground (T4), or 0.5m above-buttress (T1-T3), tree height was measured post-harvest using a surveyor's tape measure, and basic wood density was represented by the mass-weighted estimate of whole-tree basic woody tissue density.'

5- I might have missed this point. Regarding the TLS approach, did you provide estimates on the uncertainty associated with using "averaged" WD as opposed to the detailed and varying WD measurements?

5.1. Would impact both approaches (TLS and allometric) in a similar magnitude?

5.2. Are errors of the order of >20% as found by Momo Takoudjou et al and Gonzales de Tango et al. ?

We believe these points were addressed in figure 6 of the original manuscript (this is now figure 7 in the revised manuscript). This figure compares the relative error in allometric- and lidar-derived estimates of AGB, when estimates are calculated using the three considered approaches to estimating above-ground basic woody tissue density: i) mass-weighted, ii) stem and iii) literature. This is then discussed in the results (line 344) and discussion sections (line 448).

Overall, mean tree-scale relative error in lidar-derived AGB_{est} was 3%, 4% and 9% when using the mass-weighted, stem and literature estimates of above-ground woody tissue density respectively. Therefore, substantially below the error observed in previous studies. As discussed from line 390 onwards, we think this is, in part, due to the differences in the quality of the terrestrial lidar data.

We have redesigned figure 7, in the hope that it now more clearly illustrates these points. We have merged the two previously separate panels into one, such that relative error in allometric- and lidar-derived estimates of AGB can be seen side-by-side.

6- I think there is a discussion missing on the topic of tree-level AGB estimates and the extrapolation to ecological meaningful scales such as the landscape- to regional-level. TLS has a great potential to derive local reference AGB and refine tree allometry (e.g. DBH-crown size, DBH-tree height). However, similarly to traditional field techniques, TLS has spatial and temporal coverage limitations.

6.1 – Could you discuss the synergies between tree-level measurements from TLS and airborne lidar scanning (ALS)? There are now techniques to massively measure tree height and crown size from ALS in the tropics (e.g. Ferraz et al 2020, 2016) that can be converted into DBH estimates using pantropical or local allometry that converts crown size into DBH Jucker et al 2017 or Figueiredo et al 2016.

6.2. What is the role of TLS to locally define or refine tree allometry on crown size to DBH or AGB that could be then used by ALS crown measurements to calculate AGB at large scale?

6.2 The fact that most of the AGB is allocated to the crown should be emphasized and it supports the fact that ALS crown measurements can have a larger role in calculating AGB baselines as well as that allometric equations that include crown size should be further developed. This is a important point for TLS or for merging TLS and ALS.

6.3. Did you compare your results with similar work on the AGB stored in the crowns from Goodman et al 2014 (30%?) ? I believe there is more works on this.

We agree that terrestrial lidar has clear spatial limitations, and there was no context on this in the introduction. We have added the following sentence to the introduction (line 82): 'Current hardware and data acquisition protocol enable high-quality data to be collected from, for example, a 1ha tropical forest stand within a week of scanning [26].'

With respect to the synergies between terrestrial lidar and airborne lidar, we are hesitant to include a discussion on this because the data collected here do not provide any novel insights. That is, the reference harvest data do not include any measurements on crown form, to validate terrestrial lidar-derived estimates of crown form. The sample size is also too small to construct any meaningful relationships describing the correlations between: D and crown form, or AGB and crown form. There is also no coincident airborne lidar data available to compare/validate terrestrial- or airborne-lidar estimates of crown form.

In the original manuscript we compared our results on the distribution of AGB between the stem and crown, with one publication that had studied a forest similar to ours (Maia Araújo et al. 1999). We found both sets of results were consistent with one another for large trees (stem diameter > 0.6m). We have now compared our results with wider data (Goodman et al. 2014 and Ploton et al. 2016). In general, the large variance observed in crown mass ratio is comparable, but in both these latter studies, the mean crown mass ratio did tend to be smaller (i.e., below 0.5). However, we are again wary of commenting much on this given our sample size.

We have updated the following sentence in the discussion (line 527): 'Both this result, and the variation itself, is consistent with previously published data for large tropical trees in forests similar to our site [11], and also somewhat in agreement with data collected from across the tropics [13,46].'

Minor:

Line 25-26: It is not clear that "lidar AGB" is calculated from converting measured tree volume along with wood density.

The abstract now includes the sentence (line 22): 'Terrestrial lidar point clouds were collected pre-harvest, on which we fitted cylinders to model woody structure, enabling retrieval of volume-derived AGB.'

Line 37: I believe (intact) old-growth forest is commonly used in tropical ecology and science.

We have changed this sentence to read (line 16): 'Here, we harvested ... in intact old-growth forest in East Amazonia, ...'

There were further instances of this in the introduction (line 145) and conclusions (line 543), which we have now corrected.

Line 22: What volume? Total tree green mass volume? Volume of the crown envelop, stem volume? Not clear.

This has been addressed by the additional sentence described two comments above (line 22).

Line 43-47: This is amazing, it gives us an instantaneous perception of the uncertainties associated with landscape, regional and global estimates. I would state it in the abstract.

The abstract now includes the sentence (line 14): 'To date, accessible peer-reviewed data are available for just 10 large tropical trees in the Amazon that have been harvested and directly measured entirely via weighing.'

Line 73-80: I agree, but TLS has high spatial limitations. What is your opinion on estimates from tree crown measurements using airborne lidar (e.g. ((Ferraz et al 2020, 2016)) that can be then converted into biomass using regional or pantropical crown-ABG allometry as proposed in (Jucker et al 2017) or (Figueiredo et al 2016) in particular for the Brazilian Amazon?

This has been addressed in the response to major point 6.

Line 73-80: Also, what is your opinion on the synergies between TLS and ALS. I guess ALS has improved ability to measure crown size (at least diameter/perimeter) of large trees compared to TLSE. Then, crowns might store the majority of the biomass as you state in the abstract.

This has been addressed in the response to major point 6.

Line 94: question #1 and #2 are basically the same question and highly correlated to #3. Summarize?

This sentence now reads (line 111): 'However, the question remains: what is the error, both random and systematic, in these estimates?'

Line 103-104: This might be out of the scope of you work, but do you think that the lack of relationship between error and scale might be due to competition factors (open forest) or a singularity of the Eucalyptus spp. Species? This is intriguing.

This is an interesting question. In an ideal scenario, there is no inherent expectation for error in a terrestrial lidar-derived estimate of AGB to be correlated with tree size. That is, the methods for estimating above-ground green woody volume (e.g., shape fitting) are invariant of tree size, and so it would not be unreasonable to expect error to be additive. In practice, it is of course possible that this assumption will not hold. For example, in dense tropical forests, it is likely the tops of tall trees are poorly sampled, relative to the tops of small trees. Then of course ecological aspects may not be consistent across scale, such as bole rot. We would speculate then, that the lack of a correlation between error and tree size in the Calders et al. (2014) study, was, in part, due to the high quality of the lidar data.

We have slightly rewritten and rearranged the introduction to incorporate these ideas. This includes the new sentences (line 105): 'The fundamental attraction of this approach, over classical allometry, is that AGB estimates are not based upon calibration data collected from other, unrelated trees, but from explicit 3D measurements of the particular tree itself. Additionally, the various processing methods (e.g., cylinder fitting) for estimating above-ground green woody volume are scale invariant, and therefore in an idealised scenario, there is no expectation for error in these estimates to be correlated with tree size; that is, error is additive.'

And line 119 now reads: 'Perhaps the most significant finding was the absence of a correlation between error and tree size, which provided the first empirical evidence to support the aforementioned hypothesis that error in these estimates is additive.'

Line 213:135: The fact of studying only 4 trees is not a limitation of the work (it's time and resources expensive work) and I would say it earlier in the text, even in the abstract.

We are short of space in the abstract, but line 45 now reads: 'This is because directly measuring the AGB of a tree is resource intensive.'

Line 138-139: I think these are quite important and relevant analyses and findings. Good you emphasize it in the abstract.

-

Line 151: old-growth and mixed species is redundant, I believe. Is it intact forest? Somewhere in the text you call it natural but it is not clear what natural means. Is it naturally intact, natural regrowth?

The term mixed species was included to highlight the absence of persistent monodominance. The forest was intact, and this has now been included the sentence (line 167): 'The site is classified as moist, ..., intact tropical forest.'

Line 159: I never heard about "felling area" in tropical ecology and management. What does that mean?

This sentence now reads (line 174): '... felling (in particular, that no large neighbouring trees would affect felling operations) ...'

The sentence on line 178 has also been updated to read: 'Nearby vegetation surrounding each tree was removed, including complete clearing of small bushes and saplings, to enable detailed non-destructive and destructive activities.'

Line 166: 3 trees are buttress trees. Do you think that might impact the result? they might support larger crowns, taller trees and higher AGB compared with no-buttress trees?

This is certainly a possibility. These are interesting questions that could be explored with a significantly larger sample size. Here, T4, the tree without the buttress, has 59% of AGB_{ref} stored in the crown (compared with 52%, 42% and 62% for T1-T3 respectively), so it had the second largest crown mass ratio. But it is difficult to read much into this given the sample size.

Line 164-line174: I agree with the notation non-destructive methods but in fact there is a significant clearing for proper measurements. It's means than a practical situation the TLS measurements would be impacted by surrounding vegetation. What would be the errors associated with that? Is there any literature on that?

This has been addressed in the response to major point 3.

Line 181: tarpaulin? Excellent, good to know!

-

Line 178-203: remarkable work.

-

Line 247-248: The TLS and the software does not estimate biomass, they estimate tree metrics that are used to estimate biomass using auxiliary WD.

This sentence now reads (line 266): 'Estimates of above-ground woody green volume were retrieved from the lidar data ...'

And the sentence on line 275 now reads: 'Estimated above-ground biomass (AGB_{est}) was calculated ...'

Figure 3. I would suggest to combine both panels by including the bars in the upper panel. This are surprising results, Goodman found about 30% only if I can recall correctly.

Figure 3 has now been revised. As suggested, the two panels have been merged into a single combined plot. A secondary y-axis is used to show reference above-ground biomass.

Figure 4. Impressive results, namely the intra-variability and the bias. I don't understand the calculation of the solid grey line by interpolation, though.

The solid grey line was intended to show the mean basic woody tissue density at each disc location, and illustrate how this varies throughout the height of each tree. We agree this was ambiguous, and have now updated the figure to include solid grey squares to denote these averages. The legend has been updated, as has the caption, which now reads: 'The mean density at each disc location is marked by a grey square, and the solid grey interpolation line links these averages throughout each tree.'

Figure 5. Does this image has added value compared to Figure 3?

Figure 3 shows the terrestrial lidar point clouds of the four trees. Figure 5 shows the quantitative structural models (cylinder models) constructed from these clouds. The intention of this figure was to provide the reader with an understanding of the resolution and goodness of these models. We have redesigned figure 5 to consist of four separate 3D plots illustrating each model, with cylinder colour corresponding to branching order. These plots are also from different viewpoints than figure 3, so this now provides the reader with additional perspectives of the trees. Hopefully it is now clearer that figure 5 is not a rehash of figure 3.

Line 353-355. Was the error correlated with crown size or DBH? Also, is the crown size (e.g., diameter) closely related with dbh compared to with tree height?

Throughout the paper, we have used stem diameter and tree height as a proxy for tree size. This was not stated in the original manuscript. The sentence on line 67 of the introduction now reads: '... AGB is not constant across tree size (i.e., variance in AGB increases with increasing stem diameter and tree height; heteroscedasticity), ...'

And the sentence on line 364 of the discussion reads: 'increasing tree size (i.e., stem diameter and tree height)'.

Lines 356-361. Exactly, means that the estimates are not biased, crucial when scaling up to landscape or regional-level. This is quite important although only based on 4 trees.

-

Lines 395-line 399. I guess you could account for some errors, such as for example the occlusion effect. For that you could have measure the trees before and after removing the surrounding vegetation. Did you consider that?

This has been addressed in the response to major point 3.

Ferraz A, Saatchi S, Longo M and Clark D 2020 Tropical tree size–frequency distributions from airborne lidar *Ecol. Appl.* 0

Ferraz A, Saatchi S, Mallet C and Meyer V 2016 Lidar detection of individual tree size in tropical forests *Remote Sens. Environ.* 183 318–33 Online: <http://dx.doi.org/10.1016/j.rse.2016.05.028>

Figueiredo E, D'Oliveira M, Braz E, de Almeida Papa D and Fearnside P M 2016 LIDAR-based estimation of bole biomass for precision management of an Amazonian forest: Comparisons of ground-based and remotely sensed estimates *Remote Sens. Environ.* 187 281–93 Online: <http://dx.doi.org/10.1016/j.rse.2016.10.026>

Goodman R, Phillips O and Baker T 2014 The importance of crown dimensions to improve tropical tree biomass estimates *Ecol. Appl.* 24 680–98 Online: <http://dx.doi.org/10.1890/13-0070.1>

Jucker T, Caspersen J, Chave J, Antin C, Barbier N, Bongers F, Dalponte M, van Ewijk K Y, Forrester D I, Haeni M, Higgins S I, Holdaway R J, Iida Y, Lorimer C, Marshall P L, Momo S, Moncrieff G R, Ploton P, Poorter L, Rahman K A, Schlund M, Sonké B, Sterck F J, Trugman A T, Usoltsev V A, Vanderwel M C, Waldner P, Wedeux B M M, Wirth C, Wöll H, Woods M, Xiang W, Zimmermann N E and Coomes D A 2017 Allometric equations for integrating remote sensing imagery into forest monitoring programmes *Glob. Chang. Biol.* 23 177–90